

# Colour unwound – disentangling colours for azimuthal asymmetries in Drell-Yan scattering

Daniël Boer[1*], Tom van Daal[2,3♭], Jonathan R. Gaunt[2,3♮†],
Tomas Kasemets[2,3♮‡] and Piet J. Mulders[2,3♣]

**1** Van Swinderen Institute for Particle Physics and Gravity, University of Groningen,
Nijenborgh 4, NL-9747 AG Groningen, The Netherlands
**2** Department of Physics and Astronomy, VU University Amsterdam, De Boelelaan 1081,
NL-1081 HV Amsterdam, The Netherlands
**3** Nikhef, Science Park 105, NL-1098 XG Amsterdam, The Netherlands

★ d.boer@rug.nl, ♭ tvdaal@nikhef.nl, ♮ jgaunt@nikhef.nl,
♮ kasemets@nikhef.nl, ♣ mulders@few.vu.nl

## Abstract

**It has been suggested that a colour-entanglement effect exists in the Drell-Yan cross section for the 'double T-odd' contributions at low transverse momentum $Q_T$, rendering the colour structure different from that predicted by the usual factorisation formula [1]. These T-odd contributions can come from the Boer-Mulders or Sivers transverse momentum dependent distribution functions. The different colour structure should be visible already at the lowest possible order that gives a contribution to the double Boer-Mulders (dBM) or double Sivers (dS) effect, that is at the level of two gluon exchanges. To discriminate between the different predictions, we compute the leading-power contribution to the low-$Q_T$ dBM cross section at the two-gluon exchange order in the context of a spectator model. The computation is performed using a method of regions analysis with Collins subtraction terms implemented. The results conform with the predictions of the factorisation formula. In the cancellation of the colour entanglement, diagrams containing the three-gluon vertex are essential. Furthermore, the Glauber region turns out to play an important role – in fact, it is possible to assign the full contribution to the dBM cross section at the given order to the region in which the two gluons have Glauber scaling. A similar disentanglement of colour is found for the dS effect.**

---

†Present address: CERN Theory Division, CH-1211 Geneva 23, Switzerland
‡Present address: PRISMA Cluster of Excellence, Johannes Gutenberg University, Staudingerweg 7, D-55099 Mainz, Germany

## 1   Introduction

Transverse momentum dependent factorisation has been derived in proton-proton collisions for Drell-Yan (DY) and other colour-singlet productions, and for semi-inclusive deep inelastic scattering (SIDIS). Recently, these derivations have also largely been extended to colour-singlet production in double-parton scattering [2, 3]. The most complete treatment of TMD factorisation (in single-parton scattering) can be found in the book "Foundations of perturbative QCD" by J. Collins [4]. Just as for collinear factorisation, it relies among other things on the identification of leading momentum regions, the use of Ward identities, deformations out of the so-called Glauber region, and summation of multiple gluon rescatterings. The latter are required for the proper definition of the transverse momentum dependent (TMD) parton distribution functions (PDFs) or fragmentation functions (FFs), which correspond to non-local operator matrix elements. As a result, the non-local operators contain path-ordered exponentials of the gluon field, which render the TMD PDFs (or TMDs for short) gauge invariant. These path-ordered exponentials are often referred to as gauge links or Wilson lines. The observation that the paths of the gauge links depend on the process was made several times in the past (e.g. [5, 6]), but that the gauge links can affect observables was a quite an unexpected insight [7] that arose from a model calculation of the Sivers asymmetry [8]. It is now understood that the gauge links track the colour flow in the process, which in the case of DY is entirely incoming (where the corresponding initial-state interactions lead to a past-pointing staple-like Wilson line) and for SIDIS is outgoing (where the final-state interactions lead to a future-pointing staple-like Wilson line). The derivation of the gauge links in the case of more than two TMDs, where the colour flow is both incoming and outgoing, has been recognised as a problem for some time now. It has been shown that in this case the gauge links cannot be disentangled, preventing the factorisation in terms of separately colour gauge invariant fac-

tors containing TMDs [9–11]. Also the inclusion of gluonic pole factors multiplying different terms does not solve the problem as this requires weighted observables that can be expressed in terms of weighted TMDs [12]. Colour entanglement hampers the prediction of for instance TMD observables in back-to-back hadron pair production in proton-proton collisions, that use TMDs extracted from DY, SIDIS, and $e^+e^-$ annihilation measurements.

To make matters worse, a recent analysis suggested that also in the DY process 'colour-entangled' contributions can arise [1], i.e. contributions that, at best, come in a factorised form with a colour factor different from that predicted by the factorisation theorem. The affected contributions involve two T-odd TMDs, such as the Boer-Mulders (BM) function [13] and the Sivers function [14, 15]. These T-odd functions are special in the sense that their existence completely depends on the presence of the (non-straight) gauge links. In the axial gauge their contribution comes from the gluon fields at light cone infinity, that are related to the so-called gluonic pole contributions in the twist-three collinear framework [16–23]. Such 'double T-odd' contributions have been considered in the literature before [24], but not for all gluon-exchange configurations. In [1] it was derived how the colour entanglement resulted in an additional colour factor, which reduces the azimuthal $\cos(2\phi)$ asymmetry that arises from the double BM (dBM) effect [25] and even changes its overall sign. The derivation linked the dBM effect to the entanglement of two quark-quark-gluon correlators in a way similar to a double twist-three contribution without realising that in the zero-momentum limit there is a larger set of diagrams that contributes, as will be explicitly shown in this paper. The $\cos(2\phi)$ asymmetry actually has been measured in various processes and is currently under active experimental investigation by the COMPASS experiment at CERN [26, 27], and the SeaQuest experiment at Fermilab [28, 29], and is planned at NICA (at JINR) [30, 31] and J-PARC [28, 32]. Since the DY colour-entanglement result is at variance with the TMD factorisation theorem and since its experimental investigation is ongoing and planned, it is therefore important to check the result in an explicit calculation. This is the objective of this paper.

We will employ a spectator model setting, which we consider sufficiently rich in structure to establish whether there is colour entanglement in DY or not. Although the spectator cannot exhibit all the intermediate states of QCD, the diagrams, the colour matrices, and the colour factors involved all appear exactly as in the analogous full QCD calculation. We will make an explicit calculation in the model up to the order at which the colour entanglement is first anticipated to appear – this is the two-gluon exchange, or $\mathcal{O}(\alpha_s^2)$ level. We will find that the sum over all diagrams leads to a disentangled result that is in agreement with the TMD factorisation theorem for DY (with past-pointing Wilson lines). As a by-product we will see that the dBM effect at this order can be entirely ascribed to the region in which both exchanged gluons have Glauber scaling, although the fact that the effect is correctly described by the factorisation formula with only TMDs and no explicit Glauber function implies that these Glauber effects can be absorbed into the TMDs. This is related to the fact that for DY all soft momenta can ultimately be deformed into the complex plane away from the Glauber region, as discussed in the original factorisation works [4, 33–35].

Our paper is organised as follows. In the next section we will discuss the definition of the BM function and its contribution to the azimuthal-angular dependent term in the DY cross section for unpolarised hadrons. Before we move on to the factorisation calculation in the model, we will first discuss in section 3 the key elements of the factorisation proof. Subsequently, in section 4 we present an explicit model calculation that shows how the 'colour-entangled' structures are precisely disentangled, yielding the well-known factorisation formula. Sections 4.1–4.4 describe the dBM contribution only, whereas in section 4.5 we comment also on the double Sivers and double unpolarised contributions. The main results are summarised in section 5, and some technical details are given in the appendices.

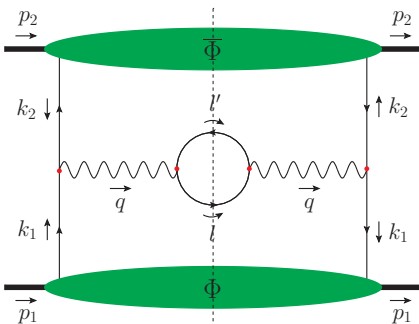

Figure 1: The DY process at leading order: a quark and antiquark are extracted from the colliding hadrons, producing a virtual photon that subsequently decays into a lepton pair. The green 'blobs' represent the quark and antiquark correlators, and the dotted line in the middle represents the final-state cut.

## 2 Extracting TMDs from observables

In this paper we focus on DY scattering, producing a virtual photon (or $Z$ boson) with momentum $q$, which in turn decays into a charged lepton-antilepton pair with momenta $l$ and $l'$. The leading-order diagram for this process is schematically illustrated in figure 1. The quark and antiquark with momenta $k_1$ and $k_2$ are extracted from the colliding hadrons (such as protons) with momenta $p_1$ and $p_2$, which is encoded by the quark and antiquark correlators $\Phi$ and $\overline{\Phi}$ respectively. These correlators can be parametrised in terms of quark and antiquark TMDs. For unpolarised protons, the quark TMD correlator can be parametrised in terms of two so-called leading-twist TMDs, namely the unpolarised function $f_1$ and the BM function $h_1^\perp$ (we will denote the antiquark analogues with a bar) [13, 36]. The quark TMDs depend on the longitudinal momentum fraction $x_1 \equiv k_1^+/p_1^+$ as well as the transverse momentum $\boldsymbol{k}_1^2$.[1]

Factorisation of DY scattering into PDFs and a perturbatively calculable hard factor was established by Collins, Soper, and Sterman (CSS) during the eighties in [34, 35], with important work in this direction also being done by Bodwin [33]. The factorisation proof for the TMD case largely proceeds along the same lines and is covered in [4]. The TMD factorisation theorem holds up to leading power in $\Lambda/Q$, where $Q^2 \equiv q^2 > 0$ represents the hard scale of the process and $\Lambda$ includes mass effects ($\sim M$), higher-twist effects ($\sim \Lambda_{\mathrm{QCD}}$) and, important for us, effects proportional to $Q_T$, where $Q_T^2 \equiv -q_T^2 = \boldsymbol{q}^2 \geq 0$ represents the non-collinearity. For unpolarised protons, the factorisation formula at leading order in the hard scattering takes the form [24, 25]:

$$\frac{d\sigma}{d\Omega\, dx_1\, dx_2\, d^2\boldsymbol{q}} = \frac{\alpha^2}{N_c\, q^2} \sum_q e_q^2 \left\{ A(\theta) \mathscr{F}\left[ f_1 \bar{f}_1 \right] + B(\theta) \cos(2\phi) \mathscr{F}\left[ w(\boldsymbol{k}_1, \boldsymbol{k}_2) h_1^\perp \bar{h}_1^\perp \right] \right\}, \quad (1)$$

with the convolution of TMDs defined as:

$$\mathscr{F}\left[ f_1 \bar{f}_1 \right] \equiv \int d^2\boldsymbol{k}_1 \int d^2\boldsymbol{k}_2\, \delta^{(2)}(\boldsymbol{k}_1 + \boldsymbol{k}_2 - \boldsymbol{q})\, f_{1,q}(x_1, \boldsymbol{k}_1^2)\, \bar{f}_{1,q}(x_2, \boldsymbol{k}_2^2). \quad (2)$$

The functions $A(\theta)$ and $B(\theta)$ are given by

$$A(\theta) = \frac{1}{4}(1 + \cos^2 \theta), \qquad B(\theta) = \frac{1}{4} \sin^2 \theta, \quad (3)$$

---

[1]Throughout the paper we make use of light-cone coordinates: we represent a four-vector $a$ as $(a^+, a^-, \boldsymbol{a})$, where $a^\pm \equiv (a^0 \pm a^3)/\sqrt{2}$ and $\boldsymbol{a} \equiv (a^1, a^2)$. We also define the four-vector $a_T$ with components $(0, 0, \boldsymbol{a})$, so that $a_T^2 = -\boldsymbol{a}^2$.

and the weight function reads

$$w(\boldsymbol{k}_1, \boldsymbol{k}_2) = \frac{2(\hat{h}\cdot\boldsymbol{k}_1)(\hat{h}\cdot\boldsymbol{k}_2) - \boldsymbol{k}_1\cdot\boldsymbol{k}_2}{M^2}. \tag{4}$$

The factorisation theorem is given in terms of the Collins-Soper angles $\theta$ and $\phi$ [37]. The unit vector $\hat{h}$ is defined in the proton centre-of-mass (CM) frame as $\hat{h} \equiv \boldsymbol{q}/|\boldsymbol{q}|$, and $p_1^2 = p_2^2 = M^2$, where $M$ is the mass of the proton. The sum in eq. (1) runs over the different quark flavours labeled by the subscript $q$. Furthermore, the electrical charge $e_q$ is given in units of the elementary charge, and $\alpha$ denotes the fine-structure constant. We refer to the first term in eq. (1) by the 'double unpolarised' contribution, because it involves unpolarised quarks. The second term describes the dBM effect, which we will focus on in this paper.

The BM function $h_1^\perp$ comes with an azimuthal-angular dependence, induced by the transverse polarisation of the quark inside the unpolarised proton. Its operator definition for a quark of flavour $q$ is given as the Fourier transform of a bilocal matrix element:

$$\frac{\widetilde{k}_{1_T}^j}{M} h_{1,q}^\perp(x_1, \boldsymbol{k}_1^2) \equiv \int \frac{d\xi^- d^2\boldsymbol{\xi}}{(2\pi)^3} e^{ik_1\cdot\xi} \langle p_1| \overline{\psi}_q(0) U_{[0,\xi]} \Gamma_T^j \psi_q(\xi) |p_1\rangle \Big|_{\xi^+=0}, \tag{5}$$

where we have employed the notation $\widetilde{a}_T^\nu \equiv \epsilon_T^{\mu\nu} a_{T\mu}$, with $\epsilon_T^{\mu\nu} \equiv \epsilon^{\mu\nu-+}$ (its non-zero components are $\epsilon_T^{12} = -\epsilon_T^{21} = 1$). A summation over colour is implicit in eq. (5) (hence the appearance of the standard $1/N_c$ colour factor in eq. (1)). Furthermore, $\Gamma_T^j$ is a Dirac projector that selects transversely polarised quarks:

$$\Gamma_T^j \equiv \frac{1}{2} i \sigma^{j+} \gamma^5, \tag{6}$$

with $j = 1,2$ and $\sigma^{\mu\nu} \equiv i[\gamma^\mu, \gamma^\nu]/2$. Eq. (5) is not in fact the full definition of the TMD – one has to accompany the bilocal matrix element by a soft factor that removes rapidity divergences and avoids double counting between the TMDs (see [4] and section 3). We do not consider this soft factor further here, however, as it will not appear in our model calculation in section 4. The definition of the BM function for the antiquark is analogous to the quark case.

The gauge link $U_{[0,\xi]}$ in eq. (5) is needed for colour gauge invariance and gives rise to a (calculable) process dependence of the TMD. For the DY process, the gauge link arises from initial-state interactions and is given by the past-pointing staple-like structure

$$U_{[0,\xi]}^{[-]} \equiv U_{[0^-,\boldsymbol{0};-\infty^-,\boldsymbol{0}]}^n U_{[-\infty^-,\boldsymbol{0};-\infty^-,\boldsymbol{\xi}]}^T U_{[-\infty^-,\boldsymbol{\xi};\xi^-,\boldsymbol{\xi}]}^n, \tag{7}$$

where the Wilson lines along the $n$ and transverse directions are given by

$$U_{[0^-,\boldsymbol{0};-\infty^-,\boldsymbol{0}]}^n \equiv \mathscr{P} \exp\left[-ig \int_0^{-\infty} d\eta^- A^+(\eta^+=0, \eta^-, \boldsymbol{\eta}=\boldsymbol{0})\right], \tag{8}$$

$$U_{[-\infty^-,\boldsymbol{0};-\infty^-,\boldsymbol{\xi}]}^T \equiv \mathscr{P} \exp\left[-ig \int_0^{\boldsymbol{\xi}} d\boldsymbol{\eta} \cdot \boldsymbol{A}(\eta^+=0, \eta^-=-\infty, \boldsymbol{\eta})\right], \tag{9}$$

and likewise for the third factor in eq. (7). For SIDIS, the gauge link arises from final-state interactions, resulting in a future-pointing link. As a consequence, the BM function is expected to change sign between DY and SIDIS [7].

It has been suggested, however, that the process dependence goes further than this sign flip. In [1] it was claimed that the dBM contribution to DY (as well as the double Sivers contribution) is suppressed and changes sign due to an additional colour factor of $-1/(N_c^2-1)$ as a result of colour entanglement. The colour-entanglement effect in [1] would signal a

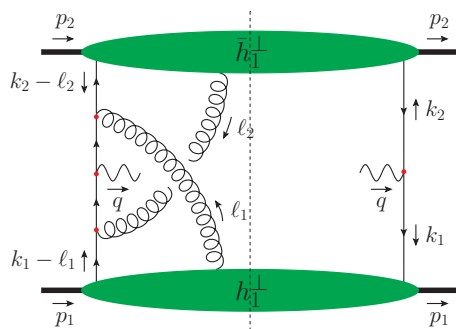

Figure 2: An example of a lowest-order graph that gives a non-zero contribution to the dBM term in the DY cross section. For convenience we have suppressed the final-state leptons.

loophole in the TMD factorisation proof of [4] for double T-odd contributions that involve polarisation. At lowest order, an entangled colour structure contributing to the dBM term for example arises in the graph in figure 2 where there is a one-gluon exchange between each correlator and the active parton coming from the other side. However, does this type of entanglement survive after summing over all relevant graphs to obtain factorisation?

In order to answer this question, we will perform an explicit factorisation calculation. To this end, we will use a spectator model that we consider rich enough in structure to settle the issue – in particular, the colour factors involved are the same as those appearing in a full QCD calculation. The calculation will be performed up to the first order at which colour entanglement is supposed to appear according to [1] – i.e. up to $\mathcal{O}(\alpha_s^2)$, thus including for example the diagram in figure 2. Before we introduce the model and present the calculation, we will remind the reader of a few key steps in the derivation of factorisation.

## 3   Approach towards factorisation

In this section we review the CSS proof for factorisation of the DY cross section at leading power [4, 33–35], focussing on the low-$Q_T$ (or TMD) contribution. A brief summary of this procedure has already been given in [2], so here we keep the presentation very compact and schematic, focussing on features that will be important in the further discussion.

The first step of the procedure is to take the possible Feynman graphs for DY production, and identify leading-power infrared regions of these diagrams – that is, small regions in the loop/phase space around the points at which certain lines go on shell, which despite being small are leading due to propagator denominators going to zero. The low-virtuality lines in these graphs are the pieces that one eventually intends to factorise off into non-perturbative functions. The infrared regions are each associated with a pinch singular surface that appears when all quantities of order $\Lambda$ in the diagram are set to zero [38, 39]. Pinch singular surfaces are surfaces where the Feynman integral contour cannot be deformed due to propagator poles 'pinching' the contour from opposite sides. The identification of pinch surfaces is aided by the Coleman-Norton theorem, which states that the pinch surfaces correspond to classically allowed processes [40].

Having determined the pinch surfaces, one needs to determine if the integration in the neighbourhood of these surfaces gives a leading contribution, and if so what the 'shape' of this leading region is. This is achieved by a power counting analysis [38, 39] – see also [4]. We choose a coordinate system in the proton CM frame where both incoming protons have zero transverse momentum, with one proton moving fast to the right and the other fast to the left. For the DY process, the power counting analysis reveals that the relevant regions of loop

momentum $\ell$ are [4, 38, 39]:

$$\text{hard } (H): \qquad \ell \sim (1, 1, 1)Q, \tag{10}$$

$$\text{right-moving collinear } (C_1): \qquad \ell \sim (1, \lambda^2, \lambda)Q, \tag{11}$$

$$\text{left-moving collinear } (C_2): \qquad \ell \sim (\lambda^2, 1, \lambda)Q, \tag{12}$$

$$\text{central soft } (S): \qquad \ell \sim (\lambda, \lambda, \lambda)Q, \tag{13}$$

$$\text{central ultrasoft } (U): \qquad \ell \sim (\lambda^2, \lambda^2, \lambda^2)Q, \tag{14}$$

$$\text{Glauber:} \qquad |\ell^+ \ell^-| \ll \ell^2 \ll Q^2, \tag{15}$$

where $\lambda$ is a small parameter which should in practice be of order $\Lambda/Q$. The soft and ultrasoft regions are treated together in the CSS methodology (see section 2.2 of [2] and references therein for more details) and in the rest of this section we use 'soft' to refer to both the soft and ultrasoft regions simultaneously. However in section 4 (and appendix A) we find it convenient to distinguish the two soft modes. The Glauber condition permits a variety of possible scalings which are all treated together in the CSS methodology. Some possible scalings, which will be important in the model analysis we perform later, are:

$$\text{right-moving Glauber } (G_1): \qquad \ell \sim (\lambda, \lambda^2, \lambda)Q, \tag{16}$$

$$\text{left-moving Glauber } (G_2): \qquad \ell \sim (\lambda^2, \lambda, \lambda)Q, \tag{17}$$

$$\text{central Glauber } (G): \qquad \ell \sim (\lambda^2, \lambda^2, \lambda)Q. \tag{18}$$

In graphs with many loops, these scalings are distributed between the loop momenta, and we have subgraphs containing lines of different scalings, which are connected via multiple lines. In the TMD case, the dominant graphs for DY have the structure shown in figure 3. There are two collinear subgraphs, one for each colliding proton. The collinear subgraph corresponding to the right-moving proton is denoted by $A$ and the other one by $B$. On both sides of the final-state cut there is a hard subgraph denoted by $H$, connected to both $A$ and $B$ by one fermion line and an arbitrary number of gluons. Lastly, there is a subgraph $S$ that initially contains both soft and Glauber partons, which connect via soft/Glauber gluon attachments to either of the collinear subgraphs.

For a region $R$ of a particular graph $\Gamma$, specified by the set of scalings for all of its loop momenta, we apply an approximator $T_R$. This approximator is appropriate to the region $R$ in the sense that within that region, $T_R\Gamma$ is equal to $\Gamma$ up to power-suppressed terms. In the computation of the contribution of each region $R$ to a graph, one actually integrates all loop momenta

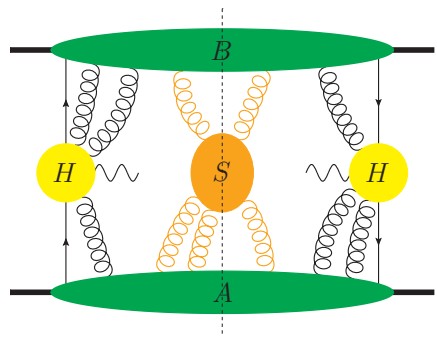

Figure 3: The partitioning of the leading DY graphs in the TMD case into various subgraphs (represented by 'blobs') that are each characterised by a particular momentum scaling. The right- and left-moving collinear subgraphs are denoted by $A$ and $B$ respectively, the soft (plus Glauber) subgraph by $S$, and the hard subgraphs by $H$.

over their full range, not only over their 'design' region. If these computations were then summed up in a naive way, then one would end up overcounting the contribution from each loop momentum region of that graph (and many of those overcounted contributions would be wrong, since their corresponding design region was different from that loop momentum region). To avoid this problem, CSS subtract terms in the computation of a region $R$, such that the final result $C_R$ for the contribution from that region is given by

$$C_R \Gamma \equiv T_R \Gamma - \sum_{R' < R} T_R C_{R'} \Gamma. \tag{19}$$

Here the integration over all loop momenta is contained in $\Gamma$. We will refer to the first and second terms on the right-hand side by 'naive graph' and 'subtraction' terms respectively. In the second term one sums over regions $R'$ whose corresponding pinch surfaces are smaller (i.e. lower dimensional) than, and lie within, that of $R$ (typically described as 'smaller regions'). With the definition (19) of the final region contribution, one can show [4] that summing over regions one obtains a correct leading-power approximation to the full graph without double counting:

$$\Gamma = \sum_R C_R \Gamma. \tag{20}$$

The presence of the subtraction terms is important in the factorisation procedure because it enables one to consider just the design region of momentum for a particular region of a graph (for example in the $A$ subgraph we can take all momenta $\ell$ to have $\ell^+ \sim Q$, and don't have to worry about when $\ell^+ \to 0$). However, one must make sure in this design region that the factorisation steps work for both the naive graph and subtraction terms, which may not always be trivial.

In the CSS proof, so-called Grammer-Yennie approximations are made for the multiple attachments of (central) soft gluon lines into the collinear subgraphs, and for the multiple attachments of unphysically-polarised collinear gluon lines into the hard subgraph. For a soft gluon momentum $\ell$ flowing out of the soft subgraph $S$ and into the right-moving collinear subgraph $A$, the form of the approximation used in [4] reads:

$$S^\mu(\ell) A_\mu(\ell) \approx S^-(\ell) A^+(\ell) = S^-(\ell) \frac{\ell^- v_R^+}{\ell^- v_R^+ + i\epsilon} A^+(\tilde{\ell}) \approx S_\mu(\ell) \frac{v_R^\mu}{\ell v_R + i\epsilon} \tilde{\ell}^\nu A_\nu(\tilde{\ell}). \tag{21}$$

In this equation, $v_R \equiv (1, -\delta^2, \mathbf{0})$ and $\tilde{\ell} \equiv (0, \ell^- - \delta^2 \ell^+, \mathbf{0})$ with $\delta$ a parameter of order $\Lambda/Q$ (the same approximation with $\delta = 0$ was used in the original CSS paper [35]). Note the appearance of $\tilde{\ell}^\nu A_\nu(\tilde{\ell})$ on the right-hand side, which is the appropriate form for the use of Ward identities. This is the utility of the Grammer-Yennie approximation – it allows us to use Ward identities to strip the soft attachments from the collinear subgraphs, after a sum over all possible soft attachments. A similar manipulation is possible for the unphysically polarised collinear attachments into the hard subgraph. If we could ignore lines with Glauber scaling, this would leave us with factorised, collinear, soft, and hard subgraphs once we also apply an appropriate projector for the physically polarised collinear-to-hard attachments. With the Grammer-Yennie approximation as in eq. (21), those soft and collinear subgraphs would contain initial-state (also referred to as past-pointing) Wilson lines.

Unfortunately, it is not possible to use the Grammer-Yennie approximation for the multiple attachments of Glauber gluons into the collinear subgraphs – one cannot neglect the transverse component of $\ell$ inside $A$ when $\ell$ is Glauber. However, in the CSS analysis of DY, it was shown that after the sum over cuts for a particular graph and region, 'final-state' poles for a Glauber momentum $\ell$ flowing into (say) $A$ cancel, leaving only 'initial-state' poles (where by 'initial-state' poles we mean poles consistent with the ultimate formation of an initial-state Wilson

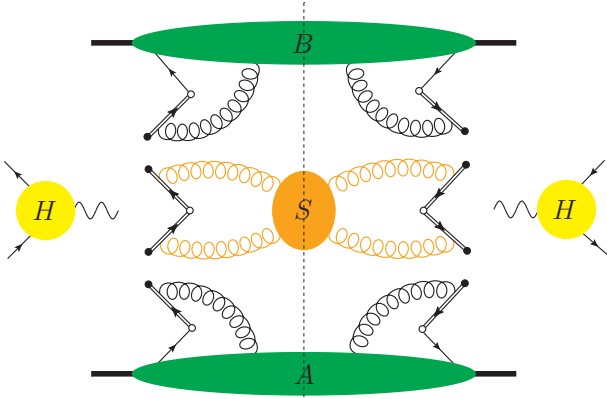

Figure 4: Factorised form of the DY process in the TMD formalism. We use the following notation for eikonal lines [3,41]: the circles at the ends of an eikonal line indicate the direction of momentum flow (from the full to the empty circle) of the original fermion, and the arrow on the line denotes the direction of colour flow (and thus also the direction of fermion number flow).

line; 'final-state' poles are on the opposite side of the complex plane from these). The physical reason underlying this cancellation is unitarity – loosely speaking, as long as the observable is insensitive to the effects of 'final-state' interactions (where here 'final-state' means that either the plus or the minus spacetime coordinate of the interaction is 'later' than that of the hard interaction), the sum over all such possible interactions gives unity (there is a unit probability for anything to happen), and the corresponding final-state poles disappear. Following the final-state pole cancellation, the integration contour for $\ell$ is no longer trapped in the Glauber region. The contours for one or both of the light-cone components of $\ell$ can be deformed into the complex plane until $\ell$ is collinear or soft, and then the Grammer-Yennie approximation (21) can be appropriately applied. Subsequently, the contours can be deformed back to the real axes again. For this final step, it is important that the Grammer-Yennie approximation does not introduce poles that obstruct the deformation back to real momenta (or if it does, the contribution from crossing these poles must not be leading power). The choice of an 'initial-state' $i\epsilon$ in eq. (21) ensures that there is no such obstruction. Effectively what happens, then, is that part of the effect of the Glauber subgraph is cancelled, and the remainder can be absorbed into the soft and/or collinear subgraphs (provided that these latter subgraphs have initial-state Wilson lines). The result of the factorisation procedure in the TMD case is shown schematically in figure 4.

The final step of the factorisation proof is the partitioning of the soft subgraph between the two collinear subgraphs, for which recently an all-order proof was provided in [42]. The result of this procedure is a factorised form with two TMDs and a hard function, as in eq. (1). In the inclusive cross section case the soft subgraph collapses down to unity, so this partitioning is trivial. The model calculation of section 4 is performed at a sufficiently low perturbative order that no soft subgraph appears that needs to be partitioned, so we will not further discuss the details here.

In the next section, our goal is to explicitly check at the two-gluon level in a model if this factorisation procedure simply works in the same way for the dBM effect, or if there are some subtleties along the lines proposed in [1]. In order to make as robust as possible a check, we will try to stick as closely as possible to a straightforward computation of the leading contribution from diagrams in the model, and then compare to the predictions of the factorisation formula (1). Graphs such as in figure 2 are complex multi-loop graphs, so a full direct evaluation with integration over all components of all loop momenta is not practical (or

indeed possible). We simplify the procedure in two ways. First, we split the calculation of the leading contribution from a diagram into a calculation of the leading regions, with appropriate approximations in each region and subtraction terms implemented as in eq. (19). Second, for each region we do not perform the integration over all loop momentum components – as we will see explicitly in section 4.3, the comparison between the predictions of the factorised formula and the explicit region calculation can already be productively done at the integrand level, with several components of several momenta unintegrated.

## 4 Model calculation

In this section we employ a spectator model in which the colourless spin-1/2 proton couples to a spin-1/2 quark and a scalar spectator (see e.g. [8,43–46]). The quark is in the triplet colour representation with electrical charge $e_q = 1$, and the scalar is in the anti-triplet colour representation and is electrically neutral. We will take the proton-quark-scalar coupling to be a constant for simplicity (as one would obtain for a fundamental Yukawa-type fermion-fermion-scalar coupling) – for convenience this vertex factor will be set to unity. The proton and scalar are taken massive with masses $M$ and $m_s$ respectively, whilst the quark is taken massless.[2] The antiproton is treated using the same spectator model as the proton, albeit with quantum numbers appropriately conjugated.

In the cross section calculations we will consider a proton colliding with an antiproton, with right- and left-collinear momenta respectively. We consider the DY production of an off-shell photon in this collision, which occurs via quark-antiquark fusion, and the scalars coupling to either hadron are spectators. To enable the hard scattering, the extracted quarks must carry right- and left-collinear momenta. In this section we will adopt momenta conventions as specified in section 2.

In the model we will consider QCD corrections to tree-level DY production. The coupling of gluons to quarks, antiquarks, and the scalar spectators is via the standard (fermionic or scalar) QCD Feynman rules. By using the standard couplings we ensure in a straightforward way that the model obeys necessary physical principles – most notably unitarity, which we will encounter in various places in the ensuing discussion.

We will for each diagram encounter a Dirac trace of the form $\text{Tr}(\Phi H_1 \overline{\Phi} H_2)$, where $H_1$ and $H_2$ represent hard scattering matrices, and $\Phi$ and $\overline{\Phi}$ are matrices for the proton and antiproton pieces respectively. Performing a Fierz transformation in Dirac space, we obtain the following decomposition [41,47]:

$$\text{Tr}\left(\Phi H_1 \overline{\Phi} H_2\right) = \text{Tr}\left(\Gamma_T^j \Phi\right) \text{Tr}\left(\overline{\Gamma}_T^k \overline{\Phi}\right) \text{Tr}\left(\overline{\Gamma}_{Tj} H_1 \Gamma_{Tk} H_2\right) + \dots, \tag{22}$$

where the Dirac projector $\Gamma_T^j$ is defined in eq. (6). We only consider the term in this sum that selects transversely polarised quarks and antiquarks, as we are interested in the dBM contribution to the differential cross section.

### 4.1 Graphs and momentum regions

To recap: our goal is to check at fixed order in a spectator model whether the dBM part of the DY cross section factorises at leading power in $\Lambda/Q$ according to eq. (1), or whether there is additional colour structure in this contribution associated with colour entanglement. We make this check at the lowest order at which a 'colour-entangled' structure is anticipated. This is the $\mathcal{O}(\alpha_s^2)$ level, which includes the diagram in figure 2. In the following, all statements are made

---

[2]To avoid issues related to proton decay, we take $m_s > M$ in our calculations.

for the dBM part of the cross section at leading power – i.e. the piece given on the right-hand side of eq. (22). Furthermore, for our calculation we adopt the Feynman gauge.

Let us first comment briefly on the single-gluon exchange, or $\mathcal{O}(\alpha_s)$ corrections. Computing the BM functions explicitly in the model (see section 4.2), one finds that the prediction of the factorisation formula is that the contribution of these to the dBM effect should be zero. At this order the only type of graph that is non-zero has a gluon extending between the scalar spectators, where this gluon has to have (central) Glauber scaling for a leading-power contribution. There are two possible places to put the final-state cut in this structure – either to the left of the Glauber gluon, or to the right – and the contributions from the two cuts exactly cancel. This cancellation is reviewed in, for example, [48].

At the $\mathcal{O}(\alpha_s^2)$ level, we find by explicit calculation that all diagrams which do not have a gluon attachment to both spectators cannot have a leading-power contribution to the dBM cross section. This leaves us with diagrams (i)–(v) in figure 5, plus graphs related to these by Hermitian conjugation or a vertical proton-antiproton flip (denoted by $p \leftrightarrow \bar{p}$),[3] and graphs which already have the colour structure anticipated by the factorisation formula (for example graph (vi) in figure 6). We also have other diagrams which only involve 'final-state' exchanges between the spectator-spectator system. The leading-power contribution from the latter class of diagrams cancels after the sum over possible final-state cuts, in an analogous way to how the one-gluon spectator-spectator exchange cancels. The contribution of diagrams (iv) and (v) plus their 'seagull' versions (i.e. those where the two gluon attachments to the lower scalar spectator leg are merged into one) also cancel after the sum over cuts, along with all 'non-colour-entangled' diagrams (except diagram (vi) and its Hermitian conjugate) – these cancellations are reviewed in appendix A.

This leaves diagram (i)–(iii) (and diagram (vi)), which we focus on in the rest of this section. For these diagrams, we identify four common non-trivial momentum regions for the gluon loop momenta $\ell_1$ and $\ell_2$ that give leading-power contributions. We use the notation $AB$ to describe the regions, where $A$ denotes the momentum scaling of the gluon with momentum $\ell_1$ and $B$ that of the gluon with momentum $\ell_2$. The four leading regions are: $G_1 G_2$, $C_1 G$, $GC_2$, and $C_1 C_2$. The region $G_1 G_2$ is the smallest one, in the sense that the pinch surface it corresponds to is the single point $\ell_1 = \ell_2 = 0$ in the eight-dimensional $\{\ell_1, \ell_2\}$-space. The regions $C_1 G$ and $GC_2$ are larger than this and overlap with one another – their pinch surfaces are lines, intersecting at the point $\ell_1 = \ell_2 = 0$. Finally, $C_1 C_2$ is the largest region, with a pinch surface that is a plane. We find that diagram (iii) only gives a leading contribution in the $GC_2$ region, so we will not consider this graph explicitly in the other regions. Likewise for the $p \leftrightarrow \bar{p}$ version of diagram (iii), which only receives a leading contribution from the $C_1 G$ region. Note that none of these regions involve a soft or ultrasoft scaling for either $\ell_1$ or $\ell_2$ – if either $\ell_1$ or $\ell_2$ is soft or ultrasoft, then the contribution to the dBM cross section from the graph is power suppressed. In the case in which $\ell_1$ or $\ell_2$ is soft, the graphs become power suppressed as too many quark lines are brought off shell to virtualities of order $\Lambda Q$ by the soft momentum. The same power suppression would also hold for these graphs in the unpolarised case. By contrast, the power suppression of the graphs when $\ell_1$ and/or $\ell_2$ is ultrasoft is specific to the spin-dependent case – here the suppression occurs in the numerator traces of eq. (22). Note that ultimately we will see that the $C_1 G$, $GC_2$, and $C_1 C_2$ regions also vanish at leading power. However, this happens in a highly non-trivial way only after the sum over graphs and possible final-state cuts, and only when the appropriate subtraction terms for smaller regions are included. Furthermore, this is related to the rapidity regulator that we use (see discussion below). Thus, we consider these regions explicitly here, detailing how and why this cancellation happens.

For each momentum region $AB$, we apply an appropriate approximator $T_{AB}$ to the graph

---

[3]For the $p \leftrightarrow \bar{p}$ versions we also flip the particle labels, i.e. $1 \leftrightarrow 2$.



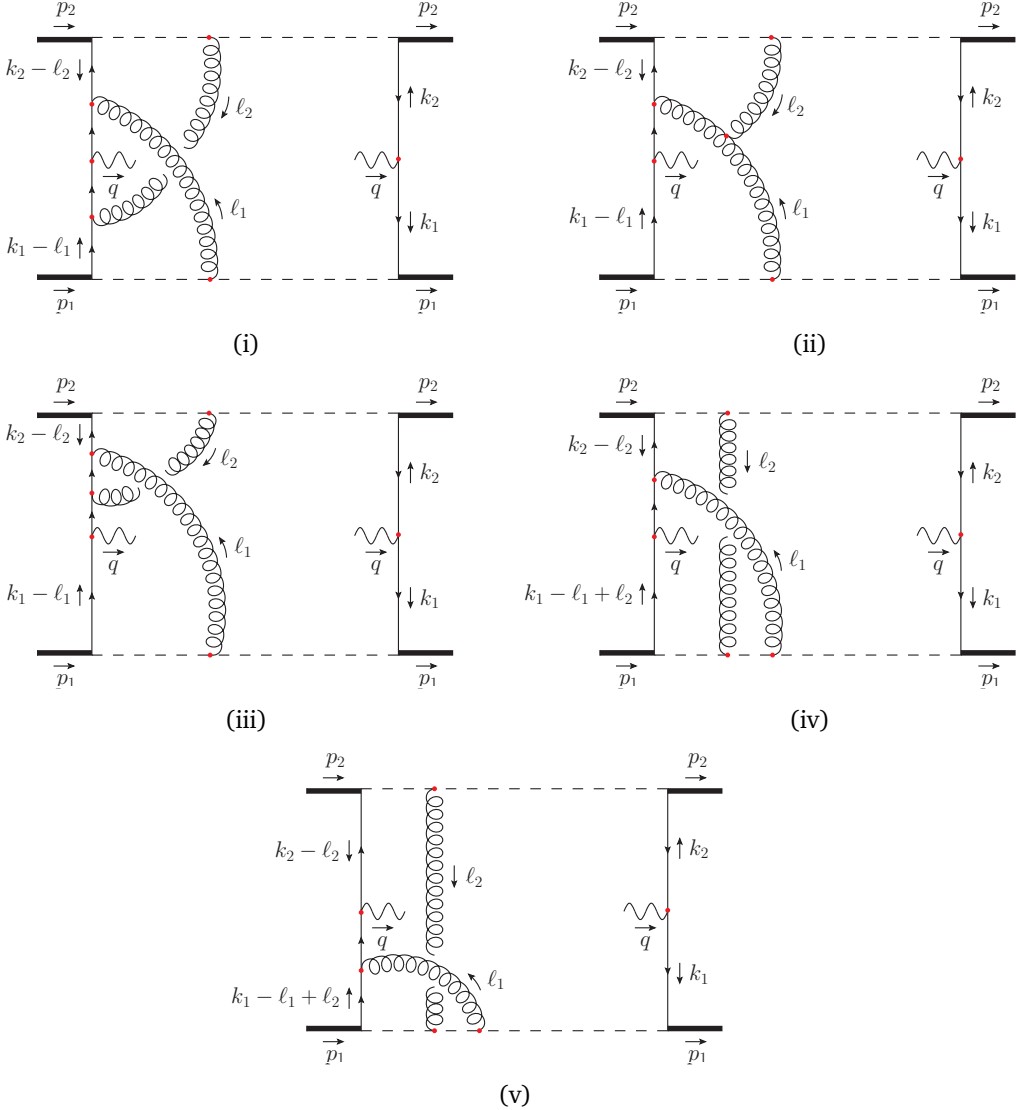

Figure 5: 'Colour-entangled' diagrams contributing to the dBM part of the DY cross section in the model at $\mathcal{O}(\alpha_s^2)$. This set is supplemented by graphs that can be obtained by $p \leftrightarrow \bar{p}$ or Hermitian conjugation, and for diagrams (iv) and (v) there are also 'seagull' versions where the two gluon attachments on the lower scalar lines merge into one.

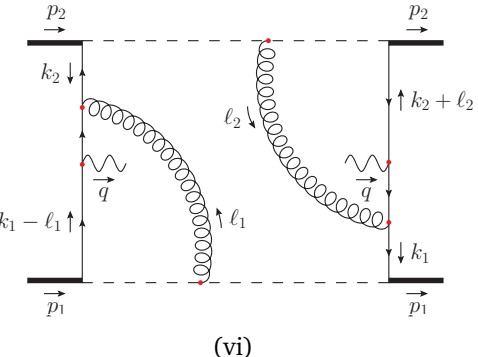

Figure 6: A diagram contributing to the dBM part of the DY cross section in the model at $\mathcal{O}(\alpha_s^2)$ that does not have a 'colour-entangled' structure.

$\Gamma$ that reduces to the unit operator (up to power corrections) in the design region $AB$. We use a 'minimal' approximator in the sense that we simply drop all terms in the numerator and propagator denominators that are power suppressed compared with other terms in the region $AB$. For the regions $G_1 G_2$, $C_1 G$, and $GC_2$, this procedure results in ill-defined results unless we also include a rapidity regulator – thus for these regions the definition of $T_{AB}$ also includes the insertion of such a regulator. The precise form of the regulator we use will be discussed below.

As prescribed by the Collins subtraction procedure, we consider the contributions from each region with subtractions from the smaller regions, according to eq. (19). To be precise, the contributions from each region are computed as follows (we omit the definition of the contribution from $GC_2$ since it is analogous to that from $C_1 G$):

$$C_{G_1 G_2} \Gamma = T_{G_1 G_2} \Gamma, \tag{23}$$

$$C_{C_1 G} \Gamma = T_{C_1 G} (1 - T_{G_1 G_2}) \Gamma, \tag{24}$$

$$C_{C_1 C_2} \Gamma = T_{C_1 C_2} (1 - T_{C_1 G} - T_{GC_2})(1 - T_{G_1 G_2}) \Gamma. \tag{25}$$

For particular graphs, one can identify further regions giving a leading-power contribution aside from the four identified above. However, the contributions from these regions can be straightforwardly absorbed into the contributions from the regions considered. For example, for diagram (i) there is also a leading-power contribution from the $GG$ region. This region overlaps with the $G_1 G_2$ region, so we should subtract out a double-counting term when considering the contribution from both regions: $T_{GG} \Gamma + T_{G_1 G_2} (1 - T_{GG}) \Gamma$. However, the only difference between the integrands of $T_{GG} \Gamma$ and $T_{G_1 G_2} T_{GG} \Gamma$ are in the propagator denominators for the active quark lines in between the gluon and hard photon vertices, and it transpires that these differences disappear after the integrations over $\ell_1^+$ and $\ell_2^-$. Hence, $T_{GG} \Gamma = T_{G_1 G_2} T_{GG} \Gamma$ and the contribution from both regions can be encapsulated by $T_{G_1 G_2} \Gamma$ (i.e. the contribution from the $GG$ region can be absorbed into the $G_1 G_2$ region).

We remark that this is the first application of the Collins subtraction procedure with the Glauber region being distinctly treated (i.e. with its own approximator, and being subtracted from larger regions). In the CSS DY factorisation proof, the Glauber and soft regions are treated together with some approximator appropriate for both, and one shows that after the cancellation of the final-state poles for the soft momenta and deformation out of the Glauber region, one can additionally apply the Grammer-Yennie approximations. Work along similar lines in which the Glauber contribution is treated distinctly and subtracted from other regions may be found in [49], although this work uses a different subtraction scheme in which the sizes of the regions are not used, and in the context of soft-collinear effective theory (SCET) in [50–52], where zero-bin subtractions [53] are used.

The regions $C_1 G$, $G_1 G_2$, and $GC_2$ are all of the same virtuality, in the sense that they have the same number of powers of the small parameter $\lambda$ in their phase space $\int d^4 \ell_1 d^4 \ell_2$ – to be specific they all have $\lambda^{10}$. They are just separated, in a sense, by rapidity. What we mean by this is particularly clear in the context of diagram (ii), where we have a gluon with momentum $\ell_1 + \ell_2$ produced by the three-gluon vertex. This gluon has the same virtuality in the $C_1 G$, $G_1 G_2$, and $GC_2$ regions, but moves in rapidity space from being $C_1$ in the $C_1 G$ region, $S$ in the $G_1 G_2$ region, and finally $C_2$ in the $GC_2$ region. The region $C_1 C_2$ then sits higher up in virtuality (the phase space has $\lambda^8$, and the gluon with momentum $\ell_1 + \ell_2$ is $H$). The relation between the regions is depicted schematically in figure 7 – we note in passing the similarity between this figure and (for example) figure 13 of [53], which depicts the momentum regions appearing in the version of SCET known as SCET$_{\mathrm{II}}$.

Since we have regions separated only by rapidity, in the computation of the contributions from these regions we must insert a rapidity regulator. For our calculations, we make a particularly simple choice that is inspired by the rapidity regulator introduced in [54, 55] – namely,

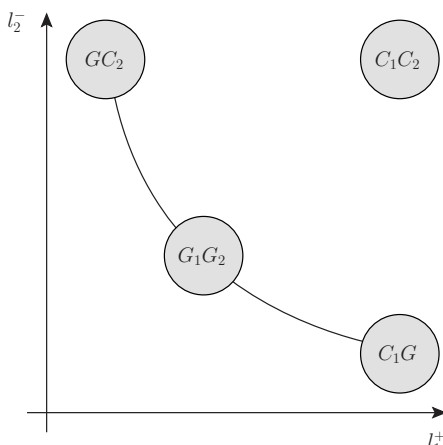

Figure 7: The relation between the relevant momentum regions for diagrams (i) and (ii). The line connecting the circles represents a surface of constant virtuality.

for all of the regions we insert the following type of regulator:

$$\left|\frac{\ell_1^+}{\nu}\right|^{-\eta_1}\left|\frac{\ell_2^-}{\nu}\right|^{-\eta_2}, \tag{26}$$

where the rapidity scale $\nu$ is a quantity with energy dimension 1 analogous to the renormalisation scale $\mu$ in dimensional regularisation, and the rapidity regulators $\eta_1, \eta_2$ are analogous to the fractional dimension $\varepsilon$ in dimensional regularisation. In the end we take the limit $\eta_i \to 0$. In fact for the three-gluon vertex graphs we have to take the limit $\eta_i \to 0$ in a particular way to obtain a well-defined result – technically, we choose slightly different regulators for the different graphs, defined as follows:

$$\text{diagram (ii):} \qquad \left|\frac{\ell_1^+}{\nu}\right|^{-\eta_1}\left|\frac{\ell_2^-}{\nu}\right|^{-\eta_2} \text{ with } \eta_1 \gg \eta_2, \tag{27}$$

$$\text{diagram (ii) with } p \leftrightarrow \bar{p}: \qquad \left|\frac{\ell_1^+}{\nu}\right|^{-\eta_{\bar{1}}}\left|\frac{\ell_2^-}{\nu}\right|^{-\eta_{\bar{2}}} \text{ with } \eta_{\bar{1}} \ll \eta_{\bar{2}}, \tag{28}$$

$$\text{diagram (i):} \qquad \frac{1}{2}\left(\left|\frac{\ell_1^+}{\nu}\right|^{-\eta_1}\left|\frac{\ell_2^-}{\nu}\right|^{-\eta_2} + \left|\frac{\ell_1^+}{\nu}\right|^{-\eta_{\bar{1}}}\left|\frac{\ell_2^-}{\nu}\right|^{-\eta_{\bar{2}}}\right). \tag{29}$$

Although having a different regulator for each graph might seem unusual, it is allowed. A full graph $\Gamma$ does not have rapidity divergences, and so does not require a rapidity regulator. According to eq. (20), this means that any rapidity regulator dependence must drop out graph by graph once we sum over all regions for that graph. This permits us to choose rapidity regulators on a graph-by-graph basis, provided that we implement subtractions for each region appropriately as in eq. (19).

The minimal requirements to get a well-defined result from diagram (ii) and its $p \leftrightarrow \bar{p}$ version are actually less restrictive than the above – one only requires $\eta_1 > \eta_2$ and $\eta_{\bar{1}} < \eta_{\bar{2}}$ – but the form above turns out to be convenient for the calculation. Similarly, for diagram (i) no hierarchy between the $\eta_i$'s actually needs to be assumed to get a well-defined answer, but the form above proves convenient. Rapidity regulators with a form different from (26) are also possible – in appendix B we discuss some alternative choices.

Note that in fact the contribution from the sum over cuts of each graph in our calculation turns out to be finite in each region when we take the appropriate $\eta_i \to 0$ limit (there are no

poles in $\eta_i$), and there is no dependence of this finite part on the quantity $\nu$. In this regard our scenario is rather different from the SCET$_{\text{II}}$ case (where $1/\eta$ divergences exist in the bare contributions from individual $C_1/S/C_2$ regions), even though the pattern of regions in figure 7 looks similar. On the other hand, our findings are consistent with other calculations involving the Glauber region – namely [51].

Now we consider the sum of diagrams (i)–(iii) (plus the $p \leftrightarrow \bar{p}$ versions of diagrams (ii) and (iii)), region by region (recall that diagram (iii) only gives a leading-power contribution in the $GC_2$ region).

**The $G_1G_2$ region.** The first region we consider is the $G_1G_2$ region. This region is in some sense the simplest to consider, and already gives some insight into the mechanics of how/whether the 'colour-entangled' structure is cancelled between the diagrams. For these reasons, we give the full details of the computation of the diagrams for this region in section 4.3.

To summarise the results for the $G_1G_2$ region: after integration over $\ell_1^{\pm}$ and $\ell_2^{\pm}$ and with the regulators as in (27)–(29), the combination of the three-gluon vertex graph, diagram (ii), with the part of diagram (i) containing the first term of (29), yields a colour structure in the sum that is consistent with factorisation. To obtain this result, it is crucial to sum over the two possible cuts of diagram (ii), one of which lies fully to the right of the gluon system, and the other of which passes through the soft gluon with momentum $\ell_1 + \ell_2$ – the sum is needed to obtain a finite result without rapidity divergences, and the result only has initial-state poles in the lower half plane for $\ell_1^+$ and $\ell_2^-$, similar to diagram (i). The $p \leftrightarrow \bar{p}$ version of diagram (ii) combines with the part of diagram (i) containing the second term of (29) to give the same result. Interestingly, the $G_1G_2$ region, at this order in $\alpha_s$ and with our chosen regulator, turns out to give the full contribution to the dBM cross section (after we also include the Hermitian conjugate diagrams, as well as diagram (vi) plus its conjugate which already have the factorised colour structure to begin with), agreeing precisely with the factorisation formula (1).

The fact that the dBM contribution comes from the 'double Glauber' $G_1G_2$ region computation fits nicely with one's expectations from the factorisation formula. As previously mentioned, the factorisation formula predicts that the dBM cross section in the model begins at $\mathcal{O}(\alpha_s^2)$, where each BM function should have one gluon attaching between the spectator and the Wilson line in the amplitude or conjugate as in figure 8. Due to the presence of an explicit factor of $i$ in the BM operator definition, the only real non-cancelled contribution to the BM function for the proton with large plus momentum is picked up when the nominally large component of the gluon momentum $\ell_1^+ \to 0$ – i.e. when the gluon goes into the Glauber region. Then, one becomes sensitive to the imaginary part of the Wilson line denominator $\ell_1^+ + i\epsilon$ and obtains a real contribution overall. Similarly for the antiproton we only obtain a real contribution when $\ell_2^- \to 0$, and the whole contribution comes only from the double Glauber region of momentum space. Bear in mind, however, that whether one obtains the dBM cross section from the $G_1G_2$ region calculation is a regulator-dependent statement, since in this calculation one integrates over the full phase space $\int d^4\ell_1 \, d^4\ell_2$ and, depending on the regulator, the integrand may do very different things outside the $G_1G_2$ design region (these differences will then be 'fixed-up' further up the subtraction hierarchy). We give examples of regulator choices for which the $G_1G_2$ region calculation does not coincide with the dBM cross section in appendix B.

Essentially, two mechanisms are responsible 'behind the scenes' for this cancellation of the colour entanglement, which are well-known and integral to the all-order proofs of factorisation in DY [4, 33–35]. The first of these is the unitarity cancellation of final-state poles after the sum over cuts of a particular diagram. This allows us to get a finite result with initial-state poles in $\ell_1^+$ and $\ell_2^-$ for the three-gluon vertex diagram after the sum over cuts (and is also responsible for the cancellation of the single-gluon exchange diagram). The second is

the non-abelian Ward identity. This ensures that when the diagrams are combined, the colour factor ends up consistent with the factorisation formula. These mechanisms will also be at play for the other regions, as we shall see.

**The $C_1G$ region.** For the $C_1G$ region, we will just consider a fixed non-zero value of $\ell_1^+$, and investigate if the 'colour-entangled' structure may be disentangled separately for the naive graph terms $T_{C_1G}\Gamma$ and subtraction pieces $T_{C_1G}T_{G_1G_2}\Gamma$. This is sufficient, since the naive and subtraction terms cancel against each other for the small $\ell_1^+$ region. We first consider the naive graph terms. The $p \leftrightarrow \bar{p}$ version of diagram (iii) vanishes upon integration over $\ell_2^-$ (and there are no subtraction terms, as this diagram is subleading in all other regions). For the $p \leftrightarrow \bar{p}$ version of diagram (ii), we find that we can write the numerator of the integrand in the following form:

$$A(\ell_1^+, k_1, k_2, T) \cdot (\ell_1 + \ell_2)^2 + B(\ell_1^+, \ell_1^-, k_1, k_2, T), \tag{30}$$

where $T$ denotes transverse variables, and $B$ contains terms that are at most linear in $\ell_1^-$. For the $B$ term, we can perform the integrations over $\ell_1^-, \ell_2^+, \ell_2^-$ (and $k_1^-, k_2^+$) using Cauchy's residue theorem (or the final-state delta functions, depending on where the position of the cut is) – for this term the fall-off in these variables is sufficiently strong at infinity such that the regulator in (28) is not needed and can be dropped. One can then show that after the sum over cuts this term vanishes – this is a unitarity cancellation of the same type as the cancellation for a single gluon mentioned above. The same procedure does not work for the $A$ term. For this piece, the integrand only falls off like one inverse power of $\ell_2^-$ at infinity (since the only factors in the denominator that depend on $\ell_2^-$ are $(\ell_1 + \ell_2)^2$ and $(k_1 + \ell_2)^2$, which depend linearly on $\ell_2^-$, and the former is cancelled by the numerator factor for the $A$ term). Then the unitarity cancellation does not work, and one needs to use the rapidity regulator (28).

Note that one might naively expect the entire contribution from the $p \leftrightarrow \bar{p}$ version of diagram (ii) to vanish in the $C_1G$ region after the sum over cuts due to unitarity arguments. This is because one has two distinct collinear systems exchanging a single Glauber gluon, so the scenario is rather similar to the $\mathcal{O}(\alpha_s)$ case (albeit with a more complex collinear system on one side) and one might expect a similar argument to work. Indeed, the contribution does vanish after the sum over cuts if one imposes physical transverse polarisations on the $C_1$ gluons (by replacing their Feynman gauge propagator numerators by axial gauge ones). The issue is that in Feynman gauge we can also have longitudinal polarisations of the $C_1$ gluons, and for these pieces the unitarity cancellation argument does not work.

For diagram (ii), one can perform a similar separation after inserting unity in the form $(\ell_2^- + i\epsilon)/(\ell_2^- + i\epsilon)$, yielding a $B$-type term that cancels after the sum over cuts of the graph, and an $A$-type term that does not. We then have $A$-type terms for the two versions of diagram (ii), where the $\ell_1 + \ell_2$ propagator has effectively been excised, plus diagram (i). These pieces are all of the same fundamental structure – for example, the only cuts possible in all of these pieces are fully to the right of the gluon system, and through the gluon with momentum $\ell_1$ (where the removal of the $\ell_1 + \ell_2$ propagator removes the possibility of an additional cut for the $A$-type terms of diagram (ii)). In fact, after the integration over $\ell_2^-$ we can combine the $A$ term of diagram (ii) with the part of diagram (i) containing the first term in (29) to yield a term with the colour factor of the factorisation formula. An analogous procedure can be done for the $A$ term of diagram (ii) with $p \leftrightarrow \bar{p}$ and the part of diagram (i) containing the second term in (29). A disentangling of the colour is then finally achieved for the naive graph terms. Note that the pattern of cancellations for the colour entanglement in these pieces is the same as for the $G_1G_2$ region.

For the subtraction terms, essentially the same techniques can be used as for the naive graph terms to disentangle the colour for $\ell_1^+ \neq 0$. Some caution is needed in taking the arguments over from the naive graph terms to the subtraction terms, owing to the fact that the

range over which $\ell_1^+$ is integrated over changes from some finite range up to values of order $Q$ in the naive graph terms, to $\pm\infty$ in the subtraction terms, and potential subtleties may exist for $|\ell_1^+| \to \infty$. We explicitly checked that with the regulators as in (27)–(29) there is no such problem, and the colour also disentangles for the subtraction terms.

Actually, since the full contribution to the factorised dBM cross section has already been accumulated in the $G_1 G_2$ region, we expect the contribution from the $C_1 G$ region not only to be colour disentangled, but to actually be zero. This is achieved once one adds the Hermitian conjugate diagrams to diagrams (i) and (ii) (diagram (vi) plus its conjugate give zero for the $C_1 G$ region).

One can treat the $G C_2$ region using an exactly analogous argument to the one just used for the $C_1 G$ region (just with $+ \leftrightarrow -$ and $1 \leftrightarrow 2$) and obtain the same result.

**The $C_1 C_2$ region.** This leaves the $C_1 C_2$ region. In this region, we can consider the naive graph terms and subtractions separately for fixed non-zero values of $\ell_1^+$ and $\ell_2^-$, due to the fact that the subtractions remove the regions where $\ell_1^+$ and/or $\ell_2^-$ are zero. When we ignore the $i\epsilon$ terms in the hard denominators (which we are allowed to do since $\ell_1^+$ and $\ell_2^-$ are non-zero), the colour between diagrams (i) and (ii) (plus the $p \leftrightarrow \bar{p}$ version of diagram (ii)) disentangles for both the naive graph and subtraction terms. This is consistent with the expectations from the non-abelian Ward identity. This procedure is explicitly worked through in the context of SCET in [56] (see also [57] where it is done in a similar fashion for one collinear and one central soft gluon). Then, combining these diagrams with diagram (vi) and all Hermitian conjugates, we obtain zero for the contribution of the $C_1 C_2$ region to the dBM cross section.

To summarise, we find for diagrams (i) and (ii) (and the $p \leftrightarrow \bar{p}$ version of diagram (ii)) that we can disentangle the colour in each of the regions $G_1 G_2$, $C_1 G$, $G C_1$, and $C_1 C_2$ separately. Once we add diagram (vi) and all Hermitian conjugate graphs, the $G_1 G_2$ region gives a result which is exactly equal to the prediction from the factorisation formula at this order, whilst the remaining regions give zero.

As mentioned in section 1, the fact that the sum over regions agrees with the factorisation formula that contains only TMDs (plus hard functions) implies that the Glauber contributions may be absorbed into these TMDs (with past-pointing Wilson lines, as appropriate for DY). The underlying reason behind this is that after the sum over cuts of diagrams (i), (ii), and (vi), one component of each gluon loop momentum (i.e. $\ell_1^+$ and $\ell_2^-$) is not trapped in the Glauber region, and $\ell_1$ may be deformed into the $C_1$ region whilst $\ell_2$ may be deformed into the $C_2$ region. This can be seen clearly in the $G_1 G_2$ region computation performed in section 4.3. The fact that the Glauber contributions can be absorbed into other region contributions is consistent with the expectations of the all-order factorisation proof [4, 33–35].

Note that by contrast, the components $\ell_1^-$ and $\ell_2^+$ are always trapped at small values of order $\Lambda^2/Q$ – they cannot be deformed after the sum over cuts of the graph, even to values of order $\Lambda$. In the case of diagram (ii), the numerator structure of the graph appears to play an important role in preventing these components from becoming untrapped after the sum over cuts of the diagram. These explicit examples show that some Glauber momenta appearing in DY cannot be deformed into central soft ones, but instead must be deformed into the collinear region – this means that the precise prescription given in section 4 of [2] of deforming all Glauber momenta into the soft region cannot be correct. The CSS works on the deformation of soft momenta out of the Glauber region [4, 34, 35] are not prescriptive about which momenta can be deformed into the collinear, and which into the central soft regions. It would be desirable to have a treatment of the Glauber modes for DY that shows in a more explicit way that all Glauber momenta can be deformed into either the central soft or collinear regions, and describes which momenta can be deformed into which region – this is however outside the scope of the present

work.

## 4.2 The Boer-Mulders function

In our diagram calculations of the dBM contribution to the DY cross section in section 4.3, we will not assume but rather derive factorisation. To be able to later identify the pieces in our $\mathcal{O}(\alpha_s^2)$ calculation that represent the quark and antiquark BM functions, we calculate $h_1^\perp$ and $\bar{h}_1^\perp$ based on the factorisation theorem. Naively, one would need to compute these up to the order at which we work, namely $\mathcal{O}(\alpha_s^2)$. However, since each function has no tree-level contribution, it suffices to compute each only to $\mathcal{O}(\alpha_s)$. The operator definition of the quark BM function is given in eq. (5). At the $\mathcal{O}(\alpha_s)$ level, the BM function is diagrammatically given in figure 8. It contains the first-order contributions to the (past-pointing) Wilson line. Since $h_1^\perp$ is a T-odd function, a gluon attachment to the eikonal line is required. There is no contribution from the case where the gluon attaches to the active quark line due to a vanishing Dirac trace, neither from graphs in which the final-state cut runs through the gluon.

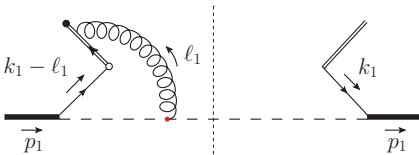

Figure 8: The first-order contribution to the quark BM function $h_1^\perp$ in our model (also the Hermitian conjugate graph is needed).

To calculate the BM function, we first identify two non-trivial momentum regions for the gluon momentum $\ell_1$ that give a leading-power contribution, namely $G_1$ and $C_1$. The $S$ and $U$ regions give power-suppressed contributions for the same reasons as discussed earlier in section 4.1. Summing over the leading regions gives, according to eqs. (19) and (20),

$$C_{G_1}\Gamma + C_{C_1}\Gamma = T_{G_1}\Gamma + T_{C_1}\left(1 - T_{G_1}\right)\Gamma. \tag{31}$$

As discussed in section 4.1 and as we will show explicitly in section 4.3, for our choice of rapidity regulators the full contribution to the dBM cross section comes from the Glauber region. Hence, only the first term in eq. (31) will turn out to be non-zero. Before applying any momentum approximations, the quark BM function is given by[4]

$$\frac{\widetilde{k}_{1r}^j}{M} h_1^\perp(x_1, \boldsymbol{k}_1^2) = -i\,C_\Phi \int \frac{d\ell_1^+}{2\pi}\,(2p_1 - 2k_1 + \ell_1)\cdot n\,\chi^j(x_1,\boldsymbol{k}_1)\frac{v^{\eta_1}|\ell_1\cdot n|^{-\eta_1}}{\ell_1\cdot n + i\epsilon} + \text{h.c.}, \tag{32}$$

where we have included the rapidity regulator $\eta_1$ (which can be send to zero at the end of our calculation), as well as the rapidity scale $v$. For convenience we suppress in this section any reference to quark flavours. Note that $h_1^\perp$ is manifestly real due to the presence of the Hermitian conjugate term (denoted by 'h.c.'). The colour factor $C_\Phi$ is given by

$$C_\Phi \equiv \text{Tr}(t^a t^a) = C_A C_F = \frac{N_c^2 - 1}{2}. \tag{33}$$

Furthermore, we have defined

$$\chi^j(x_1,\boldsymbol{k}_1) \equiv \pi g^2 \int \frac{dk_1^-}{(2\pi)^4}\,\theta[(p_1 - k_1)^0]\,\delta[(p_1 - k_1)^2 - m_s^2]\int \frac{d\ell_1^-}{2\pi}\int \frac{d^2\boldsymbol{\ell}_1}{(2\pi)^2}$$

---

[4]The necessary Feynman rules for eikonal lines are given in [3,41]. Furthermore, we make use of two light-like vectors $n^\mu \equiv (0,1,\boldsymbol{0})$ and $\bar{n}^\mu \equiv (1,0,\boldsymbol{0})$.

$$\times \frac{D_1^j}{[(k_1-\ell_1)^2+i\epsilon][(p_1-k_1+\ell_1)^2-m_s^2+i\epsilon][\ell_1^2+i\epsilon][k_1^2-i\epsilon]}, \tag{34}$$

where $\theta$ is the Heaviside step function, and $D_1^j$ is a Dirac trace given by

$$D_1^j \equiv \mathrm{Tr}\left[\Gamma_T^j(\slashed{k}_1-\slashed{\ell}_1)(\slashed{p}_1+M)\slashed{k}_1\right] = 2iM\left(x_1 p_1^+ \widetilde{\ell}_{1T}^j - \ell_1^+ \widetilde{k}_{1T}^j\right). \tag{35}$$

Let us first calculate the contribution from the $G_1$ region. We expand $h_1^\perp$ up to leading power in $\lambda$ and subsequently perform the integrals over $k_1^-$ and $\ell_1^-$. The delta function $\delta[(p_1-k_1)^2-m_s^2]$ is used for the integration over $k_1^-$ and for the integration over $\ell_1^-$ we invoke Cauchy's residue theorem. To leading power, the BM function is given by

$$\frac{\widetilde{k}_{1T}^j}{M}h_1^\perp(x_1,\boldsymbol{k}_1^2) = -2i\,C_\Phi\,(1-x_1)\,p_1^+\,\chi^j(x_1,\boldsymbol{k}_1)\int\frac{d\ell_1^+}{2\pi}\frac{v^{\eta_1}|\ell_1^+|^{-\eta_1}}{\ell_1^++i\epsilon}+\mathrm{h.c.}, \tag{36}$$

where, using the shorthand notation $\Lambda_1^2 \equiv x_1 m_s^2 - x_1(1-x_1)M^2$,

$$\chi^j(x_1,\boldsymbol{k}_1)=\frac{ig^2}{64\pi^3}\int\frac{d^2\boldsymbol{\ell}_1}{(2\pi)^2}\frac{\theta(x_1)\theta(1-x_1)D_1^j}{(p_1^+)^2[(\boldsymbol{k}_1-\boldsymbol{\ell}_1)^2+\Lambda_1^2](\boldsymbol{k}_1^2+\Lambda_1^2)\boldsymbol{\ell}_1^2}, \tag{37}$$

and

$$D_1^j = 2iM x_1 p_1^+ \widetilde{\ell}_{1T}^j. \tag{38}$$

Performing the remaining integrations gives a result for $h_1^\perp$ in the scalar spectator model that is consistent with [58,59]. Inasmuch as the function $\chi^j$ is real, only the imaginary part of the $\ell_1^+$ integral contributes to $h_1^\perp$ as its real part is canceled by the Hermitian conjugate term. This imaginary part comes from the region where $\ell_1^+$ is sensitive to the $i\epsilon$ term in the denominator, which is the case when $\ell_1^+ \to 0$ – i.e. when $\ell_1$ has Glauber scaling. Note that similar arguments were used in [8,60] to obtain single-spin asymmetries.

What happens for the $C_1$ momentum region? Here, it is sufficient to consider what happens at a fixed non-zero value of $\ell_1^+$ (the small $\ell_1^+$ region is suppressed by the subtraction). We consider the subtraction and naive graph terms, $T_{C_1}T_{G_1}\Gamma$ and $T_{C_1}\Gamma$, separately at this non-zero $\ell_1^+$. The contribution to $h_1^\perp$ from the subtraction term is also given by eq. (36), which vanishes at finite $\ell_1^+$ due to the cancellation between amplitude and conjugate. The naive graph term has a slightly different form for $\chi^j$ with respect to (37) – however this is also real-valued at non-zero $\ell_1^+$, such that amplitude and conjugate contributions cancel there too. Hence, the $C_1$ region does not contribute to the BM function, and the full contribution comes from the $G_1$ region only.

In the same way we can obtain the BM function for the antiquark, where the full contribution comes from the $G_2$ region. Since the kinematical setup is invariant under the simultaneous interchange of plus and minus indices and the particle labels 1 and 2, $\bar{h}_1^\perp$ is simply obtained from $h_1^\perp$ by the two substitutions $+ \to -$ and $1 \to 2$.

### 4.3 Calculation of the diagrams

At the order $\mathcal{O}(\alpha_s^2)$ level there are various graphs that can potentially contribute to the dBM cross section, see figures 5 and 6. In section 4.1 it was argued that some of these graphs vanish or are power suppressed, and that in all regions the sum over graphs and cuts gives zero except the $G_1G_2$ region. Here we present the explicit calculation of all 'colour-entangled' graphs and cuts for this region – namely the three diagrams (a)–(c) that are given in figure 9. These diagrams represent all possible final-state cuts for graphs (i) and (ii) given in figure 5.[5]

---

[5]Final-state cuts through Glauber gluons are not permitted as Glaubers only appear as virtual momentum modes.

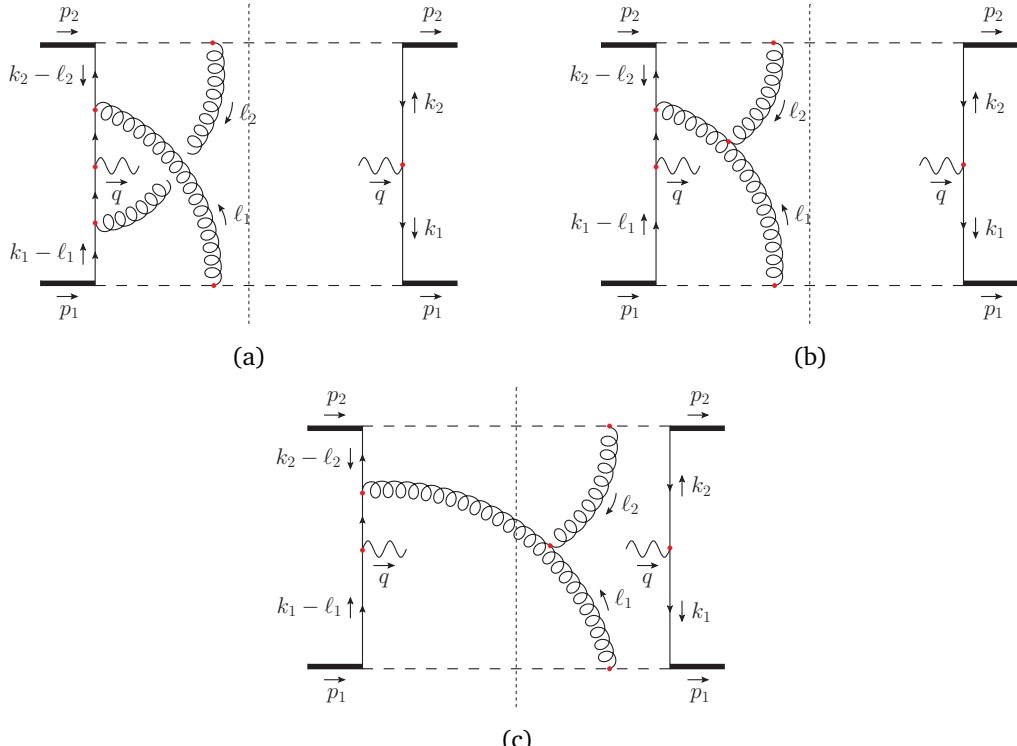

Figure 9: The relevant double-gluon exchange graphs that contribute to the dBM part of the DY cross section. This set is supplemented by graphs that can be obtained by $p \leftrightarrow \bar{p}$ or Hermitian conjugation.

The sum over all diagrams in the $G_1 G_2$ region can be written as follows:

$$
\frac{d\sigma_{\text{dBM}}}{d\Omega\, dx_i\, d^2\boldsymbol{q}} = 2\left[\frac{1}{2}\left(\frac{d\sigma_{\text{dBM}}}{d\Omega\, dx_i\, d^2\boldsymbol{q}}\right)_{(a)} + \left(\frac{d\sigma_{\text{dBM}}}{d\Omega\, dx_i\, d^2\boldsymbol{q}}\right)_{(b)}\right.
$$
$$
\left. + \left(\frac{d\sigma_{\text{dBM}}}{d\Omega\, dx_i\, d^2\boldsymbol{q}}\right)_{(c)} + \frac{1}{2}\left(\frac{d\sigma_{\text{dBM}}}{d\Omega\, dx_i\, d^2\boldsymbol{q}}\right)_{(d)} + \text{h.c.}\right], \tag{39}
$$

where $dx_i$ is short for $dx_1 dx_2$. The factor of 2 in front arises from taking into account the graphs that can be obtained from (b) and (c) (and their Hermitian conjugates) by $p \leftrightarrow \bar{p}$. Diagram (d) is given in figure 10. Although this graph already comes with the expected $1/N_c$ colour factor and will not play any role in disentangling the colour structures of diagrams (a)–(c), it is needed to obtain the full contribution from the $G_1 G_2$ region.

We now proceed with the leading-power calculation of the dBM contributions from diagrams (a)–(c) to the differential cross section in eq. (1). As mentioned, we consider only the $G_1 G_2$ region. In the following, all diagrams will be expressed in terms of the functions $\chi$ (defined in eq. (34) and simplified in eq. (37)) and $\overline{\chi}$ (the antiquark analogue of $\chi$). In this subsection, flavour labels (and the sum over different flavours) will be implicit.

**Diagram (a).** Using the decomposition in eq. (22) to select transversely polarised quarks and antiquarks, we find that the dBM contribution from diagram (a) to the differential cross section is given by

$$
\frac{1}{2}\left(\frac{d\sigma_{\text{dBM}}}{d\Omega\, dx_i\, d^2\boldsymbol{q}}\right)_{(a)} = -\frac{1}{16}\frac{\alpha^2}{q^4}\, e^2\, C_{(a)}\int d^2\boldsymbol{k}_1 \int \frac{d\ell_1^+}{2\pi}\, \chi^j(x_1, \boldsymbol{k}_1)\int d^2\boldsymbol{k}_2 \int \frac{d\ell_2^-}{2\pi}\, \overline{\chi}^k(x_2, \boldsymbol{k}_2)
$$

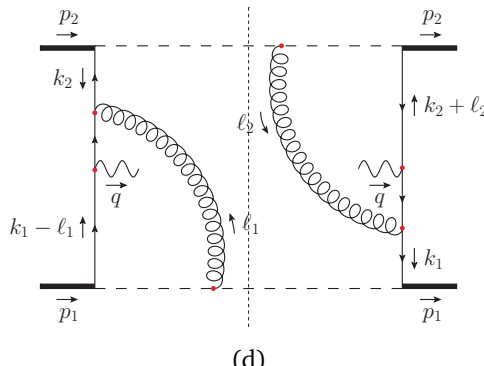

(d)

Figure 10: This diagram does not have an entangled colour structure, but is needed in the sum over diagrams to obtain the full contribution.

$$\times \frac{v^{\eta_1+\eta_2}|\ell_1^+|^{-\eta_1}|\ell_2^-|^{-\eta_2} R_{(\mathrm{a})jk;\mu\nu} L^{\mu\nu}}{[(k_2-\ell_2+\ell_1)^2+i\epsilon][(k_1-\ell_1+\ell_2)^2+i\epsilon]} \delta^{(2)}(\boldsymbol{k}_1+\boldsymbol{k}_2-\boldsymbol{q}), \quad (40)$$

where only the first term of the regulator defined in (29) appears. As explained in section 4.1, the other 'half' of diagram (a) comes with the second term in (29) and is ultimately to be combined with the $p \leftrightarrow \bar{p}$ versions of diagrams (b) and (c). The colour factor $C_{(\mathrm{a})}$ is given by

$$C_{(\mathrm{a})} \equiv \mathrm{Tr}\left(t^a t^b t^a t^b\right) = -\frac{C_F}{2} = \frac{1-N_c^2}{4N_c}, \quad (41)$$

and the Dirac trace $R_{(\mathrm{a})jk;\mu\nu}$ is defined as

$$R_{(\mathrm{a})jk;\mu\nu} \equiv \mathrm{Tr}\big[\overline{\Gamma}_{Tj}\gamma_\mu \Gamma_{Tk}(2\not{p}_1-2\not{k}_1+\not{\ell}_1)(\not{k}_2-\not{\ell}_2+\not{\ell}_1) \\ \times \gamma_\nu(\not{k}_1-\not{\ell}_1+\not{\ell}_2)(2\not{p}_2-2\not{k}_2+\not{\ell}_2)\big]. \quad (42)$$

The spin-averaged leptonic tensor $L^{\mu\nu}$ is given by

$$L^{\mu\nu} \equiv \mathrm{Tr}\left(\not{l}\gamma^\mu \not{l}'\gamma^\nu\right) = 4\left(l^\mu l'^\nu + l^\nu l'^\mu - l\cdot l'\, g^{\mu\nu}\right), \quad (43)$$

and its contraction with $R_{(\mathrm{a})jk;\mu\nu}$ reads up to leading power

$$R_{(\mathrm{a})jk;\mu\nu} L^{\mu\nu} = -128\, H_{jk}\, x_1(1-x_1)x_2(1-x_2)(p_1^+ p_2^-)^2. \quad (44)$$

Here

$$H_{jk} \equiv l_{Tj} l'_{Tk} + l_{Tk} l'_{Tj} + \boldsymbol{l}\cdot\boldsymbol{l}'\, g_{Tjk}, \quad (45)$$

where $g_T^{\mu\nu} \equiv g^{\mu\nu} - \bar{n}^\mu n^\nu - \bar{n}^\nu n^\mu$ (its non-zero components are $g_T^{11} = g_T^{22} = -1$).

To leading power, eq. (40) becomes

$$\left(\frac{d\sigma_{\mathrm{dBM}}}{d\Omega\, dx_i\, d^2\boldsymbol{q}}\right)_{(\mathrm{a})} = 2\frac{\alpha^2}{q^4} e^2 C_{(\mathrm{a})}(1-x_1)(1-x_2)p_1^+ p_2^- \int d^2\boldsymbol{k}_1\, \chi^j(x_1,\boldsymbol{k}_1)$$

$$\times \int d^2\boldsymbol{k}_2\, \overline{\chi}^k(x_2,\boldsymbol{k}_2) I_{(\mathrm{a})} H_{jk}\, \delta^{(2)}(\boldsymbol{k}_1+\boldsymbol{k}_2-\boldsymbol{q}), \quad (46)$$

where $I_{(\mathrm{a})}$ is an integral over $\ell_1^+$ and $\ell_2^-$, given by

$$I_{(\mathrm{a})} \equiv \int \frac{d\ell_1^+}{2\pi} \frac{v^{\eta_1}|\ell_1^+|^{-\eta_1}}{\ell_1^+ + i\epsilon} \int \frac{d\ell_2^-}{2\pi} \frac{v^{\eta_2}|\ell_2^-|^{-\eta_2}}{\ell_2^- + i\epsilon}. \quad (47)$$

Note that the integrand has initial-state poles in both $\ell_1^+$ and $\ell_2^-$.

**Diagram (b).** The dBM contribution from diagram (b) is given by

$$\left(\frac{d\sigma_{\text{dBM}}}{d\Omega\, dx_i\, d^2\boldsymbol{q}}\right)_{\text{(b)}} = -\frac{1}{8}\frac{\alpha^2}{q^4}\,e^2\,C_{\text{(b)}}\int d^2\boldsymbol{k}_1\int\frac{d\ell_1^+}{2\pi}\,\chi^j(x_1,\boldsymbol{k}_1)\int d^2\boldsymbol{k}_2\int\frac{d\ell_2^-}{2\pi}\,\overline{\chi}^k(x_2,\boldsymbol{k}_2)$$
$$\times\frac{v^{\eta_1+\eta_2}|\ell_1^+|^{-\eta_1}|\ell_2^-|^{-\eta_2}R_{\text{(b)}jk;\mu\nu}L^{\mu\nu}}{[(k_2+\ell_1)^2+i\epsilon][(\ell_1+\ell_2)^2+i\epsilon]}\,\delta^{(2)}(\boldsymbol{k}_1+\boldsymbol{k}_2-\boldsymbol{q}),\tag{48}$$

The colour factor $C_{\text{(b)}}$ reads

$$C_{\text{(b)}}\equiv\text{Tr}\left(if^{abc}t^a t^b t^c\right) = -\frac{C_A^2 C_F}{2} = \frac{N_c(1-N_c^2)}{4},\tag{49}$$

and the Dirac trace $R_{\text{(b)}jk;\mu\nu}$ is defined as

$$R_{\text{(b)}jk;\mu\nu}\equiv\text{Tr}\left\{\overline{\Gamma}_{Tj}\gamma_\mu\Gamma_{Tk}\left[(2p_1-2k_1+\ell_1)\cdot(2p_2-2k_2+\ell_2)(\ell_1-\ell_2)\right.\right.$$
$$+(2p_1-2k_1+\ell_1)\cdot(\ell_1+2\ell_2)(2\not{p}_2-2\not{k}_2+\not{\ell}_2)$$
$$\left.\left.-(2p_2-2k_2+\ell_2)\cdot(2\ell_1+\ell_2)(2\not{p}_1-2\not{k}_1+\not{\ell}_1)\right](\not{k}_2+\not{\ell}_1)\gamma_\nu\right\}.\tag{50}$$

Its contraction with the leptonic tensor is up to leading power given by

$$R_{\text{(b)}jk;\mu\nu}L^{\mu\nu} = 64 H_{jk}(1-x_1)x_2(1-x_2)p_1^+(p_2^-)^2\ell_1^+.\tag{51}$$

To leading power, eq. (48) becomes

$$\left(\frac{d\sigma_{\text{dBM}}}{d\Omega\, dx_i\, d^2\boldsymbol{q}}\right)_{\text{(b)}} = -2\frac{\alpha^2}{q^4}\,e^2\,C_{\text{(b)}}(1-x_1)(1-x_2)p_1^+ p_2^-\int d^2\boldsymbol{k}_1\,\chi^j(x_1,\boldsymbol{k}_1)$$
$$\times\int d^2\boldsymbol{k}_2\,\overline{\chi}^k(x_2,\boldsymbol{k}_2)\,I_{\text{(b)}}\,H_{jk}\,\delta^{(2)}(\boldsymbol{k}_1+\boldsymbol{k}_2-\boldsymbol{q}),\tag{52}$$

where $I_{\text{(b)}}$ is an integral over $\ell_1^+$ and $\ell_2^-$, given by[6]

$$I_{\text{(b)}}\equiv\int\frac{d\ell_1^+}{2\pi}\frac{v^{\eta_1}|\ell_1^+|^{-\eta_1}}{\ell_1^+ + i\epsilon}\int\frac{d\ell_2^-}{2\pi}\frac{2\ell_1^+ v^{\eta_2}|\ell_2^-|^{-\eta_2}}{2\ell_1^+\ell_2^- - (\ell_1+\ell_2)^2+i\epsilon}.\tag{53}$$

Note that the integrand has an initial-state pole in $\ell_1^+$, and, depending on the sign of $\ell_1^+$, the pole in $\ell_2^-$ is either an initial- or a final-state one.

**Diagram (c).** The dBM contribution from diagram (c) is given by

$$\left(\frac{d\sigma_{\text{dBM}}}{d\Omega\, dx_i\, d^2\boldsymbol{q}}\right)_{\text{(c)}} = -\frac{\pi^2}{8}\frac{\alpha^2}{q^4}\,e^2 g^4\,C_{\text{(b)}}\int\frac{dk_1^-}{(2\pi)^4}\int d^2\boldsymbol{k}_1\int\frac{d^4\ell_1}{(2\pi)^4}$$
$$\times\theta[(p_1-k_1+\ell_1)^0]\delta[(p_1-k_1+\ell_1)^2-m_s^2]$$
$$\times\frac{D_1^j}{[(k_1-\ell_1)^2+i\epsilon][(p_1-k_1)^2-m_s^2-i\epsilon][\ell_1^2-i\epsilon][k_1^2-i\epsilon]}$$

---

[6]Note that the factor of $\ell_1^+$ in the numerator can be understood as arising because the three-gluon $G_1 G_2 S$ system in diagram (b) is essentially the non-Wilson line part of a Lipatov vertex, and we contract the index at the end of the soft gluon line with a light-like vector travelling in the direction of $p_2$ (since the soft gluon line attaches to a line with $C_2$ scaling). This produces a numerator factor of $\ell_1^+$, as one can straightforwardly verify using the expression for the non-Wilson line part of the Lipatov vertex (see for example eq. (15) of [61]).

$$\times \int \frac{dk_2^+}{(2\pi)^4} \int d^2\boldsymbol{k}_2 \int \frac{d^4\ell_2}{(2\pi)^4} \, \theta[(p_2-k_2+\ell_2)^0] \, \delta[(p_2-k_2+\ell_2)^2-m_s^2]$$

$$\times \frac{D_2^k}{[(k_2-\ell_2)^2+i\epsilon][(p_2-k_2)^2-m_s^2-i\epsilon][\ell_2^2-i\epsilon][k_2^2-i\epsilon]}$$

$$\times \frac{v^{\eta_1}|\ell_1^+|^{-\eta_1} R_{(b)jk;\mu\nu} L^{\mu\nu}}{(k_2+\ell_1)^2+i\epsilon} \cdot 2\pi i \, \theta[-(\ell_1+\ell_2)^0] \, \delta[(\ell_1+\ell_2)^2] \, v^{\eta_2}|\ell_2^-|^{-\eta_2}$$

$$\times \delta^{(2)}(\boldsymbol{k}_1+\boldsymbol{k}_2-\boldsymbol{q}). \tag{54}$$

Note that diagrams (b) and (c) have the same colour factor.

We now expand eq. (54) to leading power in $\lambda$ and perform the integrals over the momentum components that have a $\lambda^2$-scaling, i.e. we integrate over $k_1^-$, $\ell_1^-$, $k_2^+$, and $\ell_2^+$. The two delta functions $\delta[(p_1-k_1+\ell_1)^2-m_s^2]$ and $\delta[(p_2-k_2+\ell_2)^2-m_s^2]$ are used for the integrations over $\ell_1^-$ and $\ell_2^+$, and for the integrations over $k_1^-$ and $k_2^+$ we invoke Cauchy's residue theorem. To leading power, eq. (54) becomes

$$\left(\frac{d\sigma_{\text{dBM}}}{d\Omega \, dx_i \, d^2\boldsymbol{q}}\right)_{(c)} = -2\frac{\alpha^2}{q^4} e^2 C_{(b)}(1-x_1)(1-x_2) p_1^+ p_2^- \int d^2\boldsymbol{k}_1 \, \chi^j(x_1, \boldsymbol{k}_1)$$

$$\times \int d^2\boldsymbol{k}_2 \, \overline{\chi}^k(x_2, \boldsymbol{k}_2) \, I_{(c)} \, H_{jk} \, \delta^{(2)}(\boldsymbol{k}_1+\boldsymbol{k}_2-\boldsymbol{q}), \tag{55}$$

where $I_{(c)}$ is an integral over $\ell_1^+$ and $\ell_2^-$, given by

$$I_{(c)} \equiv 4\pi i \int \frac{d\ell_1^+}{2\pi} \frac{v^{\eta_1}|\ell_1^+|^{-\eta_1}}{\ell_1^+ + i\epsilon} \int \frac{d\ell_2^-}{2\pi} \, \theta(-\ell_1^+) \ell_1^+ \, \delta[2\ell_1^+\ell_2^- - (\boldsymbol{\ell}_1+\boldsymbol{\ell}_2)^2] \, v^{\eta_2}|\ell_2^-|^{-\eta_2}. \tag{56}$$

Note that the integrand has an initial-state pole in $\ell_1^+$ and both an initial- and a final-state pole in $\ell_2^-$. The latter can be seen from the identity

$$2\pi i \, \delta(x) = \frac{1}{x-i\epsilon} - \frac{1}{x+i\epsilon}. \tag{57}$$

## 4.4 Sum of the diagrams

Let us now combine the results from diagrams (a)–(d) using eq. (39) to obtain the full dBM contribution to the DY cross section. Employing eqs. (46), (52), (55), and that $C_{(b)} = N_c^2 C_{(a)}$, gives to leading power

$$\frac{d\sigma_{\text{dBM}}}{d\Omega \, dx_i \, d^2\boldsymbol{q}} = 4\frac{\alpha^2}{q^4} e^2 C_{(a)}(1-x_1)(1-x_2) p_1^+ p_2^- \int d^2\boldsymbol{k}_1 \, \chi^j(x_1, \boldsymbol{k}_1) \int d^2\boldsymbol{k}_2 \, \overline{\chi}^k(x_2, \boldsymbol{k}_2)$$

$$\times \left[ I_{(a)} - N_c^2 \left( I_{(b)} + I_{(c)} \right) \right] H_{jk} \, \delta^{(2)}(\boldsymbol{k}_1+\boldsymbol{k}_2-\boldsymbol{q}) + \ldots + \text{h.c.}, \tag{58}$$

where the dots refer to the contribution from diagram (d) which we have not considered explicitly.

Let us now have a closer look at the integrals $I_{(b)}$ and $I_{(c)}$. Performing the integrations over $\ell_2^-$ and expanding in the regulator $\eta_2$ following our regulator prescription in (27), we obtain

$$I_{(b)} = \frac{i}{2} \int \frac{d\ell_1^+}{2\pi} \left[ \theta(-\ell_1^+) - \theta(\ell_1^+) \right] \frac{v^{\eta_1}|\ell_1^+|^{-\eta_1}}{\ell_1^+ + i\epsilon} + \mathcal{O}(\eta_2), \tag{59}$$

$$I_{(c)} = -i \int \frac{d\ell_1^+}{2\pi} \, \theta(-\ell_1^+) \frac{v^{\eta_1}|\ell_1^+|^{-\eta_1}}{\ell_1^+ + i\epsilon} + \mathcal{O}(\eta_2). \tag{60}$$

Both $I_{(b)}$ and $I_{(c)}$ have poles in $\eta_1$, which disappear once they are summed together:[7]

$$I_{(b)} + I_{(c)} = -\frac{i}{2} \int \frac{d\ell_1^+}{2\pi} \frac{\nu^{\eta_1} |\ell_1^+|^{-\eta_1}}{\ell_1^+ + i\epsilon} + \mathcal{O}(\eta_2) = I_{(a)}. \tag{61}$$

As expected, summing over all allowed cuts of graph (ii) in figure 5, i.e. adding up diagrams (b) and (c), cancels out the final-state poles present in $I_{(b)}$ and $I_{(c)}$. We are now left with $I_{(a)}$ that only has initial-state poles, consistent with the formation of initial-state Wilson lines. This cancellation of final-state poles is a consequence of the unitarity property of our model. From the integral identification (61), it follows that

$$C_{(a)} \left[ I_{(a)} - N_c^2 \left( I_{(b)} + I_{(c)} \right) \right] = \frac{1}{N_c} C_\Phi^2 I_{(a)}, \tag{62}$$

consistent with the expectations based on the non-abelian Ward identity. Now we have a colour-disentangled result from diagrams (a)–(c), which gives an expression consistent with taking the dBM part of eq. (1) and inserting the first term of eq. (36) for the proton BM function, as well as the analogous term for the antiproton BM function. The Hermitian conjugates of (a)–(c) give an expression consistent with inserting the second term of eq. (36) for the proton, and the analogue for the antiproton. Diagram (d) directly gives a result consistent with using the first term of eq. (36) for the proton, and the second term of the analogous expression for the antiproton, with the conjugate of (d) giving the other 'cross term'. The full factorised result after summing all contributions is given by:[8]

$$\frac{d\sigma_{\text{dBM}}}{d\Omega \, dx_1 dx_2 \, d^2\boldsymbol{q}} = \frac{\alpha^2}{N_c \, q^4} \sum_q e_q^2 \int d^2\boldsymbol{k}_1 \frac{\widetilde{k}_{1T}^j}{M} h_{1,q}^\perp(x_1, \boldsymbol{k}_1^2) \int d^2\boldsymbol{k}_2 \frac{\widetilde{k}_{2T}^k}{M} \bar{h}_{1,q}^\perp(x_2, \boldsymbol{k}_2^2)$$

$$\times H_{jk} \, \delta^{(2)}(\boldsymbol{k}_1 + \boldsymbol{k}_2 - \boldsymbol{q})$$

$$= \frac{\alpha^2}{N_c \, q^2} \sum_q e_q^2 \, B(\theta) \cos(2\phi) \, \mathscr{F}\left[ w(\boldsymbol{k}_1, \boldsymbol{k}_2) h_1^\perp \bar{h}_1^\perp \right]. \tag{63}$$

Although we have worked in the proton CM frame, we have expressed our final result in terms of the usual Collins-Soper angles [63]. For completeness, we have included the summation over different quark flavours.

We can now compare the result of our model calculation to the factorisation theorem in eq. (1). Since the contribution from other regions is zero, the full $\mathcal{O}(\alpha_s^2)$ result from the model is given by the $G_1 G_2$ region result in eq. (63). We can conclude that the dBM cross section precisely factorises as already anticipated by the CSS works; no loophole in their original proof for this double T-odd contribution is found. We have explicitly demonstrated that the entangled colour structures are completely disentangled after summing up the relevant diagrams. In contrast to the findings in [1], we do not find an additional colour factor on top of the standard $1/N_c$ one. In [1], diagrams (b) and (c) are not taken into account, in which case the total colour factor is simply given by

$$C_{(a)} = -\frac{1}{N_c^2 - 1} \frac{1}{N_c} C_\Phi^2. \tag{64}$$

This is precisely the colour factor that appears on the last line of eq. (10) in [1].

---

[7]The importance of combining graphs (b) and (c) in the Glauber region was also noted in [33], although there the emphasis was on performing the combination in order to be able to deform the integration contour out of the Glauber region, and the explicit discussion was only performed for the unpolarised cross section and with scalar particles such that there was no numerator structure.

[8]Since $\bar{h}_{1,q}^\perp$ belongs to a left-moving hadron rather than a right-moving one, it comes with a minus sign compared to $h_{1,q}^\perp$ from swapping plus and minus indices in the antisymmetric Levi-Civita tensor (see e.g. [62]).

## 4.5   Other observables

It is straightforward to extend the analysis just performed to certain other observables in DY for $Q_T \ll Q$. The Sivers function $f_{1T}^{\perp}$ measures the correlation between the transverse spin of a hadron and the transverse momentum of an extracted parton – in [1] it was also anticipated that there should be a colour-entanglement effect in the double Sivers (dS) part of the DY cross section obtained with polarised proton beams [64]. Similarly to the dBM part of the cross section, our spectator model also gives a non-zero result for the dS part at $\mathcal{O}(\alpha_s^2)$, coming from the same diagrams as are discussed in section 4.1. Moreover, the quark BM and Sivers functions have been shown to be identical in this model [58, 59]. Just as for the dBM case, we find here no colour entanglement, with the disentanglement being achieved via the same steps and mechanisms as given in section 4.1. The full dS cross section also resides in the $G_1 G_2$ region contribution when the regulators are chosen according to (27)–(29).

Let us also briefly remark on the case of the double unpolarised contribution to the DY cross section, for which no colour-entanglement effect is expected to appear in the final factorised result. In this case there are many more diagrams contributing at $\mathcal{O}(\alpha_s^2)$ than we considered for the dBM case, for example with more connections to the active quark/antiquark lines and less to the scalar spectator lines (in this situation gluons coupling to spectators are not strictly necessary). The computation of the factorised prediction up to $\mathcal{O}(\alpha_s^2)$ is also not so straightforward, owing to the fact that each TMD already receives contributions at $\mathcal{O}(1)$ (not to mention the appearance of the 'usual' rapidity divergences in each TMD). However, there is a part of the unpolarised cross section where the colour disentangles in essentially analogous fashion to the polarised cross sections above. In particular, note that the cancellation of the colour entanglement between diagrams (a)–(c) in the $G_1 G_2$ region proceeds in the same way for the unpolarised cross section as for the polarised cross sections. Of course, we expect that after calculating all diagrams we will fully recover the prediction from the factorisation formula, with no colour entanglement – however an explicit demonstration of this in the model is not the aim of this paper.

There exist other polarised TMDs aside from the BM and Sivers functions [65]. Of those, the function $h_{1L}^{\perp}$ is of particular interest, which describes transversely polarised quarks in a longitudinally polarised target. No colour entanglement is expected for this T-even TMD, however colour entanglement for $h_1^{\perp}$ would hamper the inclusion of the combination $h_{1L}^{\perp} + i\, h_1^{\perp}$ as a complex entry in the density matrix. This has been used to establish bounds [66] and study scale evolution [67] for these functions. As for the unpolarised case, an analysis of $h_{1L}^{\perp}$ requires the consideration of more diagrams [45], which we do not pursue here.

## 5   Conclusions

In this paper we have tested whether the lowest-order contribution to the $\cos(2\phi)$ azimuthal asymmetry in the low-$Q_T$ unpolarised DY cross section, i.e. the dBM contribution, has the usual colour structure predicted by the factorisation formula (1), or whether it has a different 'colour-entangled' structure as anticipated by [1]. This was done in the context of a spectator model. We have computed the leading-power contribution of all two-gluon exchange diagrams to the dBM cross section, by splitting each diagram into its leading-region contributions, applying approximations for those regions, and removing double counting between regions using the subtraction formalism of [4]. We did not perform any contour deformations, computing the graphs in a straightforward way in each region. For the important graphs, the relevant regions either involved both gluon momenta having Glauber scaling ($G_1 G_2$), one having Glauber and the other having collinear scaling ($C_1 G$ and $G C_2$), or both having collinear scaling ($C_1 C_2$). For the region computations to give a well-defined result, use of a rapidity regulator is needed –

we tried several possibilities, with our preferred choice given in (27)–(29).

We find that after summing over all diagrams and regions, the lowest-order predictions of the factorisation formula are precisely recovered, and the 'colour-entangled' structure anticipated by [1] cancels. The graph containing the three-gluon vertex, diagram (ii) in figure 5, summed over cuts, plays a vital role in this cancellation. In a different context, the importance of the three-gluon vertex to establish factorisation was also noted in [6]. With the regulators as in (27)–(29), the cancellation of the colour entanglement occurs on a region-by-region basis, and the full two-gluon exchange contribution to the dBM cross section ends up in the $G_1 G_2$ region. With different regulators one can shift contributions between regions – but the final result obtained from the sum over graphs and regions remains the same, as it must.

In the calculation, it is possible to identify the mechanisms that drive the cancellation of the colour entanglement. The first of these is a unitarity cancellation of the final-state poles after the sum over possible cuts of the graph. The second is the non-abelian Ward identity. Since these are rather general QCD/QFT principles, we do not expect the cancellation of colour entanglement to be restricted to the model studied, but rather to hold generally in QCD. Note that these principles were the essential ones appealed to by CSS in the original derivations of factorisation for DY.

The fact that the full calculation including contributions from the Glauber region agrees with the predictions of the factorisation formula that includes only TMDs (plus hard functions) implies that the Glauber contributions may be absorbed into these TMDs. This is related to the fact that for DY all soft momenta can ultimately be deformed into the complex plane away from the Glauber region, as discussed in [4, 33–35].

We also find a disentangling of the colour at the two-gluon level in the model for the dS contribution to the low-$Q_T$ DY cross section with transversely polarised incoming protons. It was anticipated in [1] that this contribution should also suffer from colour entanglement. The cancellation of the colour entanglement for the dS effect proceeds in a fully analogous way to that found for the dBM effect.

Of course, from our calculation we cannot say anything concrete about the possible appearance of 'colour-entangled' structures at higher orders than the one studied. However, we have no reason to doubt that the unitarity cancellation and non-abelian Ward identity drive the cancellation of the colour entanglement for the low-$Q_T$ dBM and dS cross sections also at higher orders, in line with the arguments of CSS.

## Acknowledgements

We would like to acknowledge useful discussions with Maarten Buffing, John Collins, and Markus Diehl. Furthermore, J.G. and T.K. acknowledge the hospitality of the Munich Institute for Astro- and Particle Physics (MIAPP) of the DFG cluster of excellence "Origin and Structure of the Universe". This research is in part supported by the European Community under the "Ideas" programme QWORK (contract no. 320389).

All figures were made using JaxoDraw [68, 69], and we made use of FORM [70, 71] and the FeynCalc package [72, 73].

## A   Cancellation of diagrams (iv) and (v)

In this appendix we consider diagrams (iv) and (v) given in figure 5 plus their seagull versions. These diagrams must ultimately give zero if the prediction from the factorisation formula is to be correct.

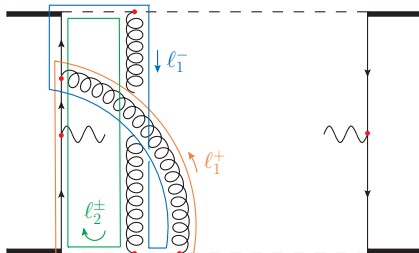

Figure 11: A convenient routing choice for the plus and minus components of $\ell_1$ and $\ell_2$ for diagram (v).

Diagram (iv) only gives a leading-power contribution when $\ell_1$ and $\ell_2$ are in the $C_1$ and $G$ regions respectively. This graph vanishes when we integrate over $\ell_1^-, \ell_2^+$, and $\ell_2^-$, and consider the sum over cuts (these integrals can be straightforwardly done using Cauchy's residue theorem). The seagull version of diagram (iv) can be cancelled by a similar argument.

Diagram (v) receives leading-power contributions from a number of momentum regions, which is possible to see if we route the plus and minus components of $\ell_1$ and $\ell_2$ as in figure 11. Using the notation from the main text where a region is denoted as $AB$, with $A$ denoting the scaling of momentum $\ell_1$ and $B$ denoting the scaling of momentum $\ell_2$, the different possible leading-power momentum regions for the momenta $\ell_1$ and $\ell_2$ in figure 11 are $UG$, $G_1G$, $C_1G$, $SG$, and $C_2G$ ($GG$ is also possible but can be absorbed into $G_1G$). Bear in mind that here the scalings of $\ell_1$ and $\ell_2$ do not precisely correspond with the scalings of the gluon momenta here, such that for the $SG$ region (for example) one gluon has $S$ scaling but the other has $G_2$ scaling. We require a rapidity regulator $\eta$ (except for the $UG$ region) for the momentum $\ell_1$ only – we make the following choices:

$$G_1G, SG : \qquad \left| \frac{\sqrt{2}\ell_1^z}{\nu} \right|^{-\eta}, \tag{65}$$

$$C_1G : \qquad \left| \frac{\ell_1^+}{\nu} \right|^{-\eta}, \tag{66}$$

$$GC_2 : \qquad \left| \frac{\ell_1^-}{\nu} \right|^{-\eta}. \tag{67}$$

We implement subtractions in each region calculation for the smaller regions according to the Collins subtraction procedure in eq. (19). With this choice of regulator, the contribution from the $G_1G$ region vanishes after the integration over $\ell_1^0$, since the two poles in this variable lie on the same side of the real axis. For each remaining region (with subtractions), the contribution cancels once we integrate over $\ell_2^\pm, k_1^-, k_2^+$, plus the appropriate component of $\ell_1$ not in the regulator function, and sum over possible cuts of the graph. For the $UG$ region one can integrate over either $\ell_1^+$ or $\ell_1^-$. The seagull version of diagram (v) is always power suppressed and can be immediately dropped.

Note that there are 'non-colour-entangled' versions of diagrams (iv) and (v) where the ordering of the two gluon attachments on the lower scalar spectator line is reversed, plus diagrams in which we have a spectator-spectator gluon exchange accompanied by an active-active gluon exchange. One must strictly speaking show that these contributions give zero to precisely validate the predictions of the factorisation formula at the given order in $\alpha_s$. The argument to show that the 'non-colour-entangled' versions of diagrams (iv) and (v) are zero proceeds in a similar fashion to that for diagrams (iv) and (v), whilst the argument for the combined spectator-spectator plus active-active gluon-exchanges diagram is rather similar to

that for the one-gluon spectator-spectator graph. We do not repeat these arguments explicitly here.

## B  Alternative rapidity regulators

The choice of rapidity regulators in (27)–(29) is not the only possible one, and different choices of regulators can result in different results for the contribution to a graph from an individual region (although the overall result after summing over regions remains the same). One other possible choice is, when $\ell_1^+$ is in the $G$ or $G_1$ region, to insert a theta function cutting off the $\ell_1^+$ integration for $|\ell_1^+| \gg Q$, and do the same for $\ell_2^-$ when it is in the $G$ or $G_2$ region (for all graphs in the same way). A possible choice for these theta functions is the following:

$$\theta(k_1^+ - \ell_1^+)\,\theta[\ell_1^+ - (p_1^+ - k_1^+)], \qquad \theta(k_2^- - \ell_2^-)\,\theta[\ell_2^- - (p_2^- - k_2^-)]. \tag{68}$$

Since the first product of theta functions naturally appears when $\ell_1$ is in the $C_1$ region, and the second appears when $\ell_2$ is in the $C_2$ region, this choice is equivalent to taking the theta functions for $\ell_1^+$ and $\ell_2^-$ appearing in the $C_1 C_2$ region, and then just using them in the smaller regions without expanding them – the approach is identical in spirit to the one taken in [49].

What is interesting in this approach is that in the $G_1 G_2$ region, diagram (c) of figure 9 with the cut running through the soft gluon gives a purely imaginary result that does not contribute to the cross section (it cancels with the Hermitian conjugate diagram). An advantage of the approach is that even in the subtraction terms $\ell_1^+$ and $\ell_2^-$ are restricted to a finite range, so the extension of arguments for the cancellation of the colour entanglement from the naive graph to the subtraction terms works in a much more straightforward way than for the method discussed in the main text. A drawback of the 'natural' choice of theta functions in (68) is that there is a proliferation of terms involving $\log[k_1^+/(p_1^+ - k_1^+)]$ and/or $\log[k_2^-/(p_2^- - k_2^-)]$ (that eventually cancel) – this can be avoided by adopting the alternative choice:

$$\theta\left[\min(k_1^+, p_1^+ - k_1^+) - |\ell_1^+|\right], \qquad \theta\left[\min(k_2^-, p_2^- - k_2^-) - |\ell_2^-|\right]. \tag{69}$$

With this choice of regulators we get the same results for diagrams (i) and (ii) in the $G_1 G_2$ region, after the sum over cuts, as one gets in section 4.3 using the regulators in (27)–(29).

Inspired by [51], one might wish to use for the $G_1 G_2$ region the regulator $|(\sqrt{2}\ell_1^z)/\nu|^{-\eta}$ $\times |(\sqrt{2}\ell_2^z)/\nu|^{-\eta}$ for diagram (i) and $|(\ell_1^+ - \ell_2^-)/\nu|^{-\eta}$ for both diagram (ii) and its $p \leftrightarrow \bar{p}$ version. In that case, diagram (i) gives the same non-zero result as in section 4.3, whilst the sum over cuts of both versions of diagram (ii) turn out to give zero – thus, with this choice of regulators the colour entanglement does not cancel region by region, but rather must cancel between regions (we recall that the sum over regions must be independent of the choice of regulator). One could even choose to use the $|(\ell_1^+ - \ell_2^-)/\nu|^{-\eta}$ regulator for diagrams (i) and (ii), in which case both of these diagrams would vanish in the $G_1 G_2$ region.

In this way we observe that by making different choices for the rapidity regulator(s), we can shift contributions between regions, and make it either more or less straightforward to demonstrate how the colour disentangles.

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
