# Peer review of "Colour unwound - disentangling colours for azimuthal asymmetries in Drell-Yan scattering"

_SciPost Physics, doi:SciPost Phys. 3, 040 (2017)_

## Round 2 · Referee Report · Anonymous · 2017-10-10

Strengths
1. A detailed analysis is given of a model calculation to test a previous published result that color entanglement effects exist in the Drell-Yan process in certain important cases. The previous result, if true, would endanger the use of standard factorization methods in these cases.
2. The analysis is sufficiently detailed to allow proper verification.
3. To support the analysis an extension of subtraction methods to the Glauber region is used. This appear to me to be new, and of importance to future work.
Weaknesses
1. There are several places where I an unable to reproduce the authors' arguments. Some appear to be on a critical path to the paper's final result. Nevertheless, I am fairly sure that they can be repaired. See the changes list for details.
2. Indications as to to the source of the error in [1] are not adequately explicit.
Report
This is an important paper. In [1] an argument was given that color entanglement effects break factorization in the Drell-Yan process, at least for the double-Boer-Mulders (dBM) or the double-Sivers (dS) cases. These are processes for which factorization proofs exist in the literature. The factorization proofs are intended to be applicable to all kinds of Drell-Yan process, with dBM and dS simply as special cases. So any failure in one case is a symptom that something is missing in the standard proofs, and hence that there may be a much more widespread issue. Given the importance of factorization to high-energy physics phenomenology, the subject is important and needs careful examination.
The present paper shows that the result in [1] is incorrect. A calculation is made at the lowest relevant order in a model, and the results do not give the color entanglement effect that the argument in [1] predicts.
As the authors say, the graphs are of 3-loop order, and hence to get useful results it is more or less essential to use appropriate approximations. In addition, the analysis can be made as much as possible without an integration over all components of loop momenta. Verification of the results depends on verifying and reproducing the chain of logic leading to the region analysis, the approximations, etc.
Unfortunately, I find that a number of steps in the argument need substantial clarification, at least.
Furthermore the paper appears to assert that the color entanglement effect found in [1] applies only to the dBM and dS situations, but not to the collinear case. This corresponds to the second paragraph of the "Discussion and conclusions" section of [1]: "One might wonder why the gauge connections cannot be disentangled as in the collinear case, since the color charges are entangled in both cases. ...". Unfortunately, I am completely unable to reproduce this result.
If it were true that the gauge connections were entangled in the collinear case, i.e., if (2) of [1] were correct, then I am totally unable to deduce a disentangled result for the collinear case. By explicit calculation, the results of entangled and disentangled connections differ at order $\alpha_s^2$, independently of whether the TMD or the collinear situation is considered. I am also only able to reproduce the formula with entanglement by omitting graphs with 3-gluon couplings.
My remarks apply equally to the earlier work JHEP 07(2011)065 by the same authors. They assert formula (4.4) with entanglement for the DY cross section, and then assert that if one of the hadrons is treated as collinear then the traces can be disentangled, to give (4.6). No proofs are given, and I have counterexamples to both (4.4) and the statement that (4.6) can be derived from (4.4).
Have I missed something?
Requested changes
1. It would be helpful if the authors could state more explicitly in the Introduction (and perhaps briefly in the Abstract) where the argument in [1] is in error. This would greatly help readers to know what issues they should focus their attention on. In the Conclusions section, it is stated that inclusion of graphs with a 3-gluon coupling is essential to the paper's result. But, on the basis of what is written I am not sure whether the failure of [1] simply lies in a failure to consider these graphs.
2. In both the Abstract and Introduction, it is stated that it is possible to assign the whole of the dBM and dS contributions to the Glauber region. However, as is explained on p. 9, the standard CSS proof shows that the Glauber contributions can also be said to cancel. There is an apparent contradiction here, and it is important that the authors point this out explicitly in the Introduction (and I think in the Abstract), and summarize how the contradiction is resolved. I get some hints about the resolution from the Conclusions, but I think it is important to bring the issue to the forefront in the Introduction, otherwise readers will be misled. I imagine that the resolution will involve two different ways of viewing the calculation.
3. In the first line of the first full paragraph on p. 3, it is stated that "a recent calculation showed that ... a color entangled contribution can arise". This needs a reference to [1] at this point. But the present paper falsifies the result of [1], at least at the lowest order, $\alpha_s^2$ at which the entanglement effect is supposed to happen. So I don't think "showed" is the correct word.
4. In addition, the experimental work referred to on p. 3 needs to have explicit references.
5. The meaning to be applied to "color-entangled" should be defined, preferably in the Introduction.
6. In the paragraph just above (1), it is stated that factorization of DY into TMDs and a hard factor was established by Bodwin and CSS. The use of the word "TMDs" indicates that TMD factorization was intended by the authors. But the referred-to papers [26-28] only claimed to prove factorization for the collinear case. The critical part of the proof concerning the Glauber region and spectator interactions does actually extend to the TMD case, but I don't know of an adequate use of it for the TMD case until [4].
7. In the definition of the BM function (5), there is a soft factor missing.
8. The notation for the Wilson lines in (7) and as used just above it needs correction, for two reasons. First, the transverse and + components of $\eta$ aren't specified. Second, the use of the arguments $a$ and $b$ as limits of a one-dimensional integral indicates that $a$ and $b$ are single numbers. But in the 3rd factor on the previous line $U$ has a $\xi$ argument is a 4-vector. Knowledgeable readers will know what is intended, but less knowledgeable readers will have trouble.
9. At the top of p. 6, it is stated that "the factorisation theorem was derived for unpolarised particles". In fact the proof, as least as given in [4], is explicitly intended to apply to all the polarised cases, and this includes the dBM and DS cases.
10. I was somewhat confused by the last paragraph on p. 6. It is perhaps worth saying more explicitly that the actual pinches of relevance only occur in the massless limit (i.e., $m/Q \to 0$); it is in this limit that the Coleman-Norton method is used. The regions of interest are neighborhoods of the pinch-singular surfaces.
11. On p. 7 a list of the relevant regions is given. Only later is it stated that Glauber-type regions are also relevant. However, before a sum over cuts, the Glauber regions are unsuppressed. I think it would be better to bring these two lists together, otherwise the categorical statement of the relevant regions is actually falsified a paragraph later. A reference for the assertions is needed, since it is quite non-trivial to justify that these are the appropriate regions and scalings, and that no other scalings need be considered.
12. It is not at all obvious to me that the soft scaling given is the appropriate one. The ultrasoft scaling is also possible. At a minimum, a reference to the discussion in [2] is needed. It might even be worth going into some detail here, since understanding the regions quantitatively is critical to the calculations. The ultrasoft region does give leading-power contributions to cut graphs for the cross section (although they do not matter, I think, for dBM and dS).
13. There is a clash in the literature between the scaling parameter $\lambda$ being treated as an external parameter, say $\Lambda_{\rm QCD}/Q$, and $\lambda$ being treated as an integration variable. The first view is predominant in the SCET literature and I think is also found in [2] and elsewhere. The second view is found in [4]. Some remarks on this might help.
14. At the end of the paragraph containing (8)-(11), it is stated that "In the inclusive DY cross section, collinear scalings with the large component pointing in arbitrary directions (corresponding to final-state jets in arbitrary directions) are also initially important, but such jets cancel after the sum over final states". While there are theorems that have some resemblance to this stated result, I am not aware of any that have this precise statement. A reference is needed. Have the authors written what they mean? In any case, for the TMD case, such configurations are actually power-suppressed.
15. At (17), the justification for the sign of the $i\epsilon$ in the denominators needs to be given or referred to, since the Glauber region is under consideration. The authors point out that the third equality in (17) fails in the Glauber region. But also the second equality is dangerous because before contours are deformed, $l^-$ is integrated through zero. To make the second equality valid, there has to be a deformation away from the pole at $l^-=0$. It is not immediately clear that this is consistent with other singularities in the $l^-$ plane. A careful discussion is needed. There is relevant material in CSS and in [4], but something needs to be said here.
16. The approximation in (17) retains the transverse component of $l$ in the $A$ factor. As already discussed in [2], this is at variance with what Collins used in [4]. The prescription in [4] appears to be important to get the Ward identity argument to work for summing over the soft attachments to a collinear factor without getting extra terms in the non-abelian case (i.e., QCD). I think some discussion is needed.
17. In the 2nd line of the 3rd paragraph of Sec. 4.1, "dBM" needs to be inserted before "cross section".
18. At the bottom of p. 11, the regions giving leading-power contributions are listed. It would be helpful to state explicitly why soft and ultrasoft gluons do not need to be considered in addition. After all the pinch surfaces are the same for both Glauber and soft momenta.
19. In Sec. 4.1, the paper introduces subtraction methods that include a treatment of the Glauber region. These appear to be new and of potentially widespread application, so they are worth mentioning in the Introduction (and possibly the Abstract). Or is there previous work along the same lines?
Tom van Daal on 2017-11-03 [id 187]
We would like to thank the referee for his/her quick response and providing valuable feedback on the manuscript. We fully agree with most requested changes. We would like to refer to version 3 for a list of comments on the implemented changes.

---

## Round 3 · Referee Report · Anonymous · 2017-11-9

Report

Generally, the authors have dealt with my concerns properly in the revised version. However, there are still a few areas where I still find problems.

1. The response to item 1, includes the following: "... the (often used) incorrect statement that the Glauber contributions cancel". I don't see that this is an incorrect statement, at least for a certain valid use of the terminology, as I will try to explain below. But there is clearly a different usage where one can say that the Glauber contributions do not cancel. The view with the cancellation is essential to deriving factorization, while the view with the non-cancellation is useful for calculations such as are made in Sec. 4.

Now, a key point in the Libby-Sterman analysis is that, in determining the infra-red regions to be considered in a graph, it is only necessary to consider those regions where the integration contour is trapped. If one deforms the contour away from regions without a pinch, contributions to a process only arise from regions corresponding to pinches. In individual cut graphs for the Drell-Yan process there are pinches in the Glauber region. One can try moving the contour away from pinches involving Glauber momenta, but only at the price of extra contributions from final-state poles that are crossed. These contributions can have Glauber momenta, but they cancel to leading power after a sum over cuts. What is left is an integral over the deformed contour, which does not go through Glauber momenta. With this view, which is correctly summarized in the paper, I think it is correct to say that the Glauber contributions cancel (as always with the proviso "to leading power").

A different point of view is also possible, which leads to the calculational methods devised in the paper with its explicit use of contributions from the Glauber region, and associated subtractions.

I think a resolution can be motivated by the situation in the Brodsky-Hwang-Schmidt (BHS) calculation of a one-gluon contribution to the Sivers asymmetry in SIDIS. To get the asymmetry an imaginary part in an amplitude is needed, and BHS calculate this by setting the appropriate intermediate state on-shell. In the calculation, the gluon is in the Glauber region, and one can therefore say that the asymmetry is given by the Glauber contribution. (Essentially, the same idea appears in calculation in Sec. 4.2 in the present paper for the BM function.)

But in an analysis to derive factorization, one can deform the integration out of the Glauber region into the collinear region for the gluon, and thereby derive factorization. The imaginary part now comes from the fact that the integral over the longitudinal momentum fraction of the gluon goes through complex values to avoid a pole. In this view, there is no Glauber contribution, only a collinear contribution.

Therefore, unless I have misunderstood something, the above argument leads to the need for the paper to contain further explanation of these issues. I am pretty sure that these different views are known and understood by the authors. But making them and their relations as fully explicit as possible would enable readers to clearly understood them as well, and would considerably assist future work.

2. Concerning the justification that the soft and ultrasoft regions do not need to be considered for the dBM cross section: The authors have added a statement making the assertion that these regions are power suppressed. I think it would be very helpful also to add a summary of how this is known, at least if this can be done briefly. A couple of sentences may suffice. In addition, I now see that some similar remarks are needed for the calculation of the BM function in Sec. 4.2. These changes would improve the reproducibility of the results. I suspect that readers who are not intimately familiar with the issues involved will probably have great difficulty verifying this result in the absence of suitable hints. They may well know that soft and/or ultrasoft contributions are generally leading (and even divergent) in the graphs considered, but may not find it obvious that the situation changes for the BM/dBM case.

3. A related issue, particularly prominent for the calculation of the BM function, is about the need for a collinear term in (31). (A generalization is also relevant for the calculation of the dBM cross section.) The final contribution to the BM function at this order is only from the Glauber region. If there is no collinear contribution, why bother with the collinear term in (31)? After all, there are no (leading-power) soft and ultrasoft contributions, so no corresponding terms are used in (31).

I think the reason is that the power suppression for the soft and ultrasoft regions in the BM function happens graph-by-graph, whereas the collinear contribution is unsuppressed for individual graphs. The collinear contribution only cancels after adding the hermitian conjugate graph, so in a graph-by-graph analysis it must be retained. In addition, the use of the collinear term is needed to allow a matching by contour deformation with the factorization point of view with its attribution of the BM function to the collinear region with complex momentum fraction. Could the authors confirm whether or not my understanding is correct? If so, it would be useful to add something to Sec. 4.2 (and/or elsewhere) to explain near the beginning why they use formulas containing collinear terms even though the final answer is only from Glauber contributions.

4. I am still rather confused by the explanation of how the derivations in Ref. [1] came to be in error. But since that issue concerns a different paper, I will not pursue it further.

5. On p. 6, it is stated that the TMD factorization proof in [4] did not explicitly include proton or quark polarization effects. It is indeed true that the chapter that treats the DY case seems to restrict its mention of these effects to inserting the words "polarization effects" in factorization formulas. However, the chapter should be read in the context of previous chapters, where polarization effects are explicitly treated. The relevant parts include (a) Sec. 11.9 for the Ward identities, especially Sec. 11.9.5 for the generalization beyond DIS (since Ch. 11 is about DIS); (b) Chs. 12 and 13. There the methods explicitly include polarization; See, for example, (13.22), where projectors on Dirac spinor space are applied that enable all leading power polarization effects to be included. The application of Ward identities for collinear gluon attachments to hard subgraphs and for soft gluon attachments to collinear subgraphs is such that they factorize independently of polarization. (Earlier chapters contain the details.)

It is surely intended to be taken for granted that the same procedures are to be applied to Drell-Yan. The extra step for DY is the demonstration of cancellation of final state terms, in Sec. 14.4. There is no need to explicitly mention polarization, since all necessary polarization dependent terms are inside the various shaded subgraphs.

Of course, it is always possible there is a mistake; the treatment of the final state poles is very non-trivial, and the dBM and dS processes are particularly sensitive to any problems. Certainly it would be nice to improve the proof. But in identifying potential sources of breakdown or improvement of the proof it is important to correctly identify what was done and not done. If after considering my remarks, the authors are still concerned that issues concerning polarization are missed, it would help for them to specify more precisely what their concern is, if that is possible.

6. In addition, I have noticed the following minor issue: In the second paragraph of Sec. 2, the references for proofs of factorization in [33-35] include the Bodwin paper, but the only authors mentioned are CSS. Either include Bodwin in the list of people or remove also CSS.

Requested changes

See report.

  • validity: -
  • significance: -
  • originality: -
  • clarity: -
  • formatting: -
  • grammar: -

Author Tom van Daal on 2017-11-21
(in reply to Report 1 on 2017-11-09)
Category:
remark
answer to question
correction

We would like to thank the referee for his/her very careful reading of the manuscript and providing valuable feedback. Below we comment on each one of the points raised and provide suggestions for changes where needed.

  1. We fully agree with the referee that one can justify the usage of “Glauber contributions cancel” in the way explained. It indeed depends on the viewpoint adopted. In our paper we touch upon both viewpoints mentioned (see the introduction, the conclusions, and especially the end of section 4.1) and, most important for our purposes, no inconsistencies are identified. In our view the discussion in the current version of our paper does not raise any questions concerning this topic. We believe that the problem lies with our response to the referee, that did not sufficiently address both viewpoints, but not with the present version of our paper, which is in our view sufficiently explicit about the two viewpoints to avoid confusion. In the conclusions we write: “The fact that the full calculation including contributions from the Glauber region agrees with the predictions of the factorisation formula that includes only TMDs (plus hard functions) implies that the Glauber contributions may be absorbed into these TMDs. This is related to the fact that for DY all soft momenta can ultimately be deformed into the complex plane away from the Glauber region, as discussed in [4,33–35].” We do not see the need for a more detailed discussion of Glauber contribution cancellation in factorization proofs and refer to the literature for that purpose. Our aim was not to write a review paper on factorization, but rather to show consistency with it. Our work concerns the study of colour factors associated to double T-odd contributions, and we conclude that those contributions factorise precisely as predicted by CSS (regardless of which viewpoint here is adopted). As a by-product, we find that the full contribution to the dBM cross section at the order considered can be ascribed to the Glauber region (which in fact depends on the rapidity regulator used, see appendix B), i.e. if no contour deformations are performed. We note that upon summing over cuts, the integrals over the Glauber components can in fact also be deformed into the collinear region, which is consistent with the arguments of CSS. We agree with the referee that indeed now the necessary imaginary part would come from the fact that the contour goes through complex values to avoid the pole. Note that the information on the pole (or the sensitivity to $i\epsilon$) remains in the direction in which you have deformed the contour.

Currently we have below eq. (38) in section 4.2:

“Inasmuch as the function $\chi^j$ is real, only the imaginary part of the $\ell_1^+$ integral contributes to $h_1^\perp$ as its real part is canceled by the Hermitian conjugate term. This imaginary part comes from the region where $\ell_1^+$ is sensitive to the $i\epsilon$ term in the denominator, which is the case when $\ell_1^+ \to 0$ – i.e. when $\ell_1$ has Glauber scaling.”

We would like to add to this the following sentence containing a reference to the BHS papers:

“Note that similar arguments were used in \cite{Brodsky:2002cx,Brodsky:2002rv} to obtain single-spin asymmetries.”

  1. Currently we have the following sentence at the end of the first complete paragraph on p.12:

“Note that none of these regions involve a soft or ultrasoft scaling for either $\ell_1$ or $\ell_2$ – if either $\ell_1$ or $\ell_2$ is soft or ultrasoft, then the contribution to the dBM cross section from the graph is power suppressed.”

We agree with the referee that a brief expansion on this statement would be helpful. Hence, we propose to add the following:

“In the case in which $\ell_1$ or $\ell_2$ is soft, the graphs become power suppressed as too many quark lines are brought off shell to virtualities of order $\Lambda Q$ by the soft momentum. The same power suppression would also hold for these graphs in the unpolarised case. By contrast, the power suppression of the graphs when $\ell_1$ and/or $\ell_2$ is ultrasoft is specific to the spin-dependent case – here the suppression occurs in the numerator traces of eq. (22).”

In section 4.2, just above eq. (31), we currently write:

“To calculate the BM function, we first identify two non-trivial momentum regions for the gluon momentum $\ell_1$ that give a leading-power contribution, namely $G_1$ and $C_1$. Summing over these regions gives, according to eqs. (19) and (20),”

We propose to change this into the following:

“To calculate the BM function, we first identify two non-trivial momentum regions for the gluon momentum $\ell_1$ that give a leading-power contribution, namely $G_1$ and $C_1$. The $S$ and $U$ regions give power-suppressed contributions for the same reasons as discussed earlier in section 4.1. Summing over the leading regions gives, according to eqs. (19) and (20),”

  1. The reason why we consider the collinear contribution explicitly and not the soft/ultrasoft contribution is precisely as the referee identifies – the soft/ultrasoft contributions vanish straightforwardly graph-by-graph (so a detailed discussion of them is not necessary), whilst the collinear contributions only turn out to vanish after a suitable sum over graphs and cuts (and, for the dBM cross section, this cancellation is very non-trivial), so we consider these explicitly and show that they do cancel, as well as carefully considering why they cancel. Furthermore, the cancellation in the collinear sector is not happening already at the ‘naive graph term’ level but only because we have the Glauber subtraction terms implemented, making it even more non-trivial and important to demonstrate explicitly. Finally, the fact that the collinear contributions vanish is related to the rapidity regulator that we use – in appendix A it is shown that choices of rapidity regulator exist for which the collinear contributions are not zero.

Given this, we suggest to include the following after the added paragraph for point (2):

“Note that ultimately we will see that the $C_1G$, $GC_2$, and $C_1C_2$ regions also vanish at leading power. However, this happens in a highly non-trivial way only after the sum over graphs and possible final-state cuts, and only when the appropriate subtraction terms for smaller regions are included. Furthermore, this is related to the rapidity regulator that we use (see discussion below). Thus, we consider these regions explicitly here, detailing how and why this cancellation happens.”

  1. Okay.

  2. We realise that we were not completely accurate here and agree with the referee. Currently, we state on p.6:

“Although the TMD factorisation proof in [4] was intended to apply to all polarised cases, no explicit proton or quark polarisation was in fact considered. The colour-entanglement effect in [1] would signal a loophole in the proof for the double T-odd contributions that involve polarisation.”

We propose to change this into the following:

“The colour-entanglement effect in [1] would signal a loophole in the TMD factorisation proof of [4] for double T-odd contributions that involve polarisation.”

Additionally, in the first sentence of the first complete paragraph on p.17, we should omit the word “unpolarised”.

Furthermore, on p.26 we currently have:

“We can conclude that the dBM cross section factorises in the same way as the double unpolarised contribution that was considered in the original CSS factorisation proof.”

We propose to change this sentence into the following:

“We can conclude that the dBM cross section precisely factorises as already anticipated by the CSS works; no loophole in their original proof for this double T-odd contribution is found.”

  1. We acknowledge that this is indeed a bit sloppy. We currently have:

“Factorisation of DY scattering into PDFs and a perturbatively calculable hard factor was established by Collins, Soper, and Sterman (CSS) during the eighties in [33–35] and fully completed in 2011 also for the TMD case in [4].”

We propose to replace this by the following:

“Factorisation of DY scattering into PDFs and a perturbatively calculable hard factor was established by Collins, Soper, and Sterman (CSS) during the eighties in [33–34], with important work in this direction also being done by Bodwin [35]. The factorisation proof for the TMD case largely proceeds along the same lines and is covered in [4].”

Furthermore, we currently have at the end of section 3:

“The final step of the factorisation proof is the partitioning of the soft subgraph between the two collinear subgraphs – once this is done one has the final factorised form with two TMDs and a hard function, as in eq. (1).”

We suggest to replace this by the following:

“The final step of the factorisation proof is the partitioning of the soft subgraph between the two collinear subgraphs, for which recently an all-order proof was provided in \cite{Vladimirov:2017ksc}. The result of this procedure is a factorised form with two TMDs and a hard function, as in eq. (1).”

---

## Round 3 · Author Response

We would like to thank the referee for his/her quick response and providing valuable feedback on the manuscript. We have revised and improved the paper accordingly. Below we comment on each one of the requested changes.

---

## Round 3 · List of Changes

1. The origin of the entanglement in [1] stems from the form of the link structure assumed (in eq. (2) and fig. 2 of [1]), which is based on rules given in a paper by Bomhof, Mulders, and Pijlman (Eur.Phys.J. C47 (2006) 147-162), which were not derived for the case of dBM or dS. For these cases, the rules would lead to a colour entanglement as in [1], just as for instance for quark-quark-gluon correlations (with non-zero gluon momentum) in the situation of double twist-three contributions, although that situation may have other problems with factorisation. It is precisely the fact that one is dealing with zero-momentum gluons that leads to a larger set of diagrams than the set accounted for in [1], as demonstrated in the present paper. We have now pointed this out in the Introduction.

2. What the CSS proof shows is the cancellation of final-state poles that prevent deformations out of the Glauber region into the soft/collinear regions. After the final-state pole cancellation, effects from the Glauber region of momentum do still remain, which may then be absorbed into the soft and collinear functions (i.e. the TMDs). This point is emphasised in section 3. Moreover, in various places in the text, we have emphasised that even though the contribution at the order considered can be viewed as coming entirely from the Glauber region, it can also be absorbed entirely into the standard BM TMDs - see the sentence beginning with “As a by-product...” in the Introduction, a similar sentence in the Conclusions, and the detailed discussion at the end of section 4.1. The apparent contradiction thus comes from the (often used) incorrect statement that Glauber \textit{contributions} cancel.

3. We have added the reference and reformulated the sentence, also incorporating point 5. It now reads: “To make matters worse, a recent analysis suggested that also in the DY process ‘colour-entangled’ contributions can arise [1], i.e. contributions that, at best, come in a factorised form with a colour factor different from that predicted by the factorisation theorem. The affected contributions involve two T-odd TMDs, such as the Boer-Mulders (BM) function [13] and the Sivers function [14,15].”

4. We have added references here.

5. See point 3.

6. We have reformulated this sentence to read: “Factorisation of DY scattering into PDFs and a perturbatively calculable hard factor was established by Collins, Soper, and Sterman (CSS) during the eighties in [33–35] and fully completed in 2011 also for the TMD case in [4].”

7. We have added the following sentence after eq. (6): “Eq. (5) is not in fact the full definition of the TMD – one has to accompany the bilocal matrix element by a soft factor that removes rapidity divergences and avoids double counting between the TMDs (see [4] and section 3). We do not consider this soft factor further here, however, as it will not appear in our model calculation in section 4.”

8. We were indeed a bit sloppy here; the definition of the Wilson line is now more explicit, see eqs. (7)–(9).

9. We have reformulated this sentence to read: “Although the TMD factorisation proof in [4] was intended to apply to all polarised cases, no explicit proton or quark polarisation was in fact considered. The colour-entanglement effect in [1] would signal a loophole in the proof for the double T-odd contributions that involve polarisation.”

10. We have modified the paragraph to state that the pinch surfaces only appear in the massless limit.

11. We fully agree that it makes more sense to bring the two lists together, and have done so. We have also added references.

12. We have added the ultrasoft scaling along with the soft scaling to the list of regions, noting that these are typically treated together in the CSS methodology and citing [2] and references therein.

13. In our paper we now write explicitly that “$\lambda$ is a small parameter which should in practice be of order $\Lambda/Q$”, hoping that no reader will assume it is an integration parameter.

14. Since the paper is entirely focussed on the TMD case, the discussion of the collinear case has simply been removed now.

15. We have changed the Grammer-Yennie approximation to the one of [4] – see point 16. We mention one key place in which the Grammer-Yennie approximation fails for the Glauber region (see the discussion at the end of p.9). At the top of p.10, we have explained why the sign of $i\epsilon$ in the denominators has been chosen as it has. We hope that this is sufficiently clear to the reader.

16. We have now altered the discussion to use the Grammer-Yennie approximation of [4]. This brief discussion of the Grammer-Yennie approximation is only there to illustrate how soft-collinear and collinear-hard attachments can be stripped away by Ward identities, but that the same procedure does not work for Glauber gluons. Eq. (17) is not used anywhere else in the paper, and in our opinion, a rather detailed technical discussion of the relative merits of the prescriptions in [2] and [4] would detract from the flow of the paper.

17. We have inserted ‘dBM’ here.

18. We have added a sentence on p.12 (at the end of the second paragraph) pointing out that if $\ell_1$ or $\ell_2$ is soft or ultrasoft in figures (i)–(iii), the contribution to the dBM cross section is power suppressed. The region where $\ell_1$ is ultrasoft and $\ell_2$ is Glauber can contribute at leading power in diagram (v), but this contribution also cancels - we have added some discussion of this in appendix B.

19. We have added a paragraph in section 4.1 noting that this paper involves the first application of the Collins subtraction method in which the Glauber region is treated in a distinct way. There is some work along similar lines that makes use of other subtraction schemes – this work is cited in the added paragraph.

---

## Editorial Decision

published